# Universality in Learning from Linear Measurements

**Ehsan Abbasi**
Department of Electrical Engineering
California Institute of Technology
Pasadena, CA, 91125
eabbasi@caltech.edu

**Fariborz Salehi**
Department of Electrical Engineering
California Institute of Technology
Pasadena, CA, 91125
fsalehi@caltech.edu

**Babak Hassibi**[*]
Department of Electrical Engineering
California Institute of Technology
Pasadena, CA, 91125
hassibi@caltech.edu

## Abstract

We study the problem of recovering a structured signal from independently and identically drawn linear measurements. A convex penalty function $f(\cdot)$ is considered which penalizes deviations from the desired structure, and signal recovery is performed by minimizing $f(\cdot)$ subject to the linear measurement constraints. The main question of interest is to determine the minimum number of measurements that is necessary and sufficient for the perfect recovery of the unknown signal with high probability. Our main result states that, under some mild conditions on $f(\cdot)$ and on the distribution from which the linear measurements are drawn, the minimum number of measurements required for perfect recovery depends only on the first and second order statistics of the measurement vectors. As a result, the required of number of measurements can be determining by studying measurement vectors that are Gaussian (and have the same mean vector and covariance matrix) for which a rich literature and comprehensive theory exists. As an application, we show that the minimum number of random quadratic measurements (also known as rank-one projections) required to recover a low rank positive semi-definite matrix is $3nr$, where $n$ is the dimension of the matrix and $r$ is its rank. As a consequence, we settle the long standing open question of determining the minimum number of measurements required for perfect signal recovery in phase retrieval using the celebrated PhaseLift algorithm, and show it to be $3n$.

## 1 Introduction

Recovering a structured signal from a set of linear observations appears in many applications in areas ranging from finance to biology, and from imaging to signal processing. More formally, the goal is to recover an unknown vector $\mathbf{x}_0 \in \mathbb{R}^n$, from observations of the form $y_i = \mathbf{a}_i^\mathsf{T} \mathbf{x}_0$, for $i = 1, \ldots, m$. In many modern applications, the ambient dimension of the signal, $n$, is often (overwhelmingly) larger than the number of observations, $m$. In such cases, there are infinitely many solutions that satisfy the linear equations arising from the observations, and therefore to obtain a unique solution one must assume some prior structure on the unknown vector. Common examples of structured signals are sparse and group-sparse vectors [13, 6], low-rank matrices [24, 5], and simultaneously-structured matrices [8, 21]. To this end, we use a convex penalty function $f : \mathbb{R}^n \to \mathbb{R}$, that captures the *structure* of the structured signal, in the sense that signals that do not adhere to the desired structure

---
[*]This work was supported in part by the National Science Foundation under grants CNS-0932428, CCF-1018927, CCF-1423663 and CCF-1409204, by a grant from Qualcomm Inc., by a grant from Futurewei Inc., by NASA's Jet Propulsion Laboratory through the President and Director's Fund, and by King Abdullah University of Science and Technology.

will have a higher cost. Therefore, the following estimator is used to recover $\mathbf{x}_0$,
$$\hat{\mathbf{x}} = \arg\min_{\mathbf{x}} \ f(\mathbf{x}) \quad \text{subject to,} \quad y_i = \mathbf{a}_i^\mathsf{T}\mathbf{x}, \ i = 1, \ldots, m \,. \tag{1}$$

Popular choices of $f(\cdot)$ include the $\ell_1$-norm for sparse vectors [31], and the nuclear norm for low-rank matrices [24]. A canonical question in this area is "how many measurements are needed to recover $\mathbf{x}_0$ via this estimator?" This question has been extensively studied in the literature (see [28, 1, 9] and the references therein.) The answer depends on the $\mathbf{a}_i$ and is very difficult to determine for any given set of measurement vectors. As a result, it is common to assume that the measurement vectors are drawn randomly from a given distribution and to ask whether the unknown vector can be recovered with high probability. In the special case where the entries of the measurement matrix are drawn iid from a Gaussian distribution, the minimum number of measurements for the recovery of $\mathbf{x}_0$ with high probability is known (and is related to the concept of the Gaussian width [28, 1, 9]). For instance, it has been shown that $2k\log(n/k)$ linear measurements is required to recover a $k-$sparse signal [12], and $3rn$ measurements suffice for the recovery of a symmetric $n \times n$ rank-$r$ matrix [20, 9]. Recently, Oymak et al [22] showed that these thresholds remain unchanged, as long as the entries of each $\mathbf{a}_i$ are *i.i.d* and drawn from a "well-behaved" distribution. It has also been shown that similar universality holds in the case of noisy measurements [23]. Although these works are of great interest, the independence assumption on the entries of the measurement vectors can be restrictive. In certain applications in communications, phase retrieval, covariance estimation, the entries of the measurement vectors $\mathbf{a}_i$ have correlations. In this paper, we show a much stronger universality result which holds for a broader class of measurement distributions. Here is an informal description of our result:

> *Assume the measurement vectors $\mathbf{a}_i$ are drawn iid from some given distribution. In other words, the measurement vectors are iid random, but their entries are not necessarily so. Then the minimum number of observations needed to recover $\mathbf{x}_0$ from (1) with high probability, depends only on the first two statistics of the $\mathbf{a}_i$, i.e., their mean vector $\mu$, and covariance matrix $\Sigma$.*

We anticipate that this universality result will have many practical ramifications. In this paper we focus on the ramifications to the problem of recovering a structured matrix, $\mathbf{X}_0 \in \mathbb{R}^{n \times n}$, from quadratic measurements (a.k.a. rank-one projections). In this problem, we are given observations of the form $y_i = \mathbf{a}_i^\mathsf{T}\mathbf{X}_0\mathbf{a}_i = \text{Tr}(\mathbf{X}_0(\mathbf{a}_i\mathbf{a}_i^\mathsf{T})) = \text{vec}(X)^t\text{vec}(\mathbf{a}_i\mathbf{a}_i^t)$ for $i = 1, \ldots, m$.[2] Such measurement schemes appear in a variety of problems [11, 3, 35, 19, 18]. An interesting application of learning from quadratic measurements is the PhaseLift algorithm [7] for phase retrieval. In phase retrieval, the goal is to recover the signal $\mathbf{x}_0$ from quadratic measurements of the form, $y_i = |\mathbf{a}_i^\mathsf{T}\mathbf{x}_0|^2 = \mathbf{a}_i^\mathsf{T}(\mathbf{x}_0\mathbf{x}_0^\mathsf{T})\mathbf{a}_i$. Note that $\mathbf{x}_0\mathbf{x}_0^t$ is a low-rank (in this case rank-1) matrix and PhaseLift relaxes this constraint to a non-negativity constraint and minimizes nuclear norm to encourage a low rank solution. Quadratic measurements also appears in non-coherent energy measurements in communications and signal processing [33, 2], sparse covariance estimation [11, 35], and sparse phase retrieval [18, 26]. Recently, Chen et al [11] proved sufficient bounds on the number of measurements for various structures on the matrix $\mathbf{X}_0$. However, to the best of our knowledge, prior to this work, the precise number of required measurements for perfect recovery was unknown.

For example, when the $\mathbf{a}_i$ have iid Gaussian entries (note that the measurement vectors, which are now $\text{vec}(\mathbf{a}_i\mathbf{a}_i^t)$, are no longer iid Gaussian) we show that $3nr$ measurement is necessary and sufficient for the perfect recovery of a rank-$r$ matrix from quadratic measurements. In the special case of phase retrieval, we therefore demonstrate that $3n$ measurements is necessary and sufficient for perfect recovery of $\mathbf{x}_0$, which settles the long standing open question of the recovery threshold for PhaseLift. In particular, this indicates that $2n$ extra phaseless measurements is all that is needed to compensate the missing phase information.

The remainder of the paper is structured as follows. The problem setup and definitions are given in Section 2. In Section 3, we introduce our universality framework, which states that the number of required observations for the recovery of an unknown model depends only on the first two statistics of the measurement vectors. As an applications, in Section 4, we apply this universality theorem to derive tight bounds (i.e., necessary and sufficient conditions) on the required number of observations for matrix recovery via quadratic measurements.

## 2 Preliminaries

### 2.1 Notations

We start by introducing some notations that are used throughout the paper. Bold lower letters $\mathbf{x}, \mathbf{y}, \ldots$ are used to denote vectors, and bold upper letters $\mathbf{X}, \mathbf{Y}, \ldots$ are for matrices. For a matrix $\mathbf{X} \in \mathbb{R}^{m \times n}$,

$\text{Vec}(\mathbf{X}) \in \mathbb{R}^{mn}$ returns the vectorized form of the matrix. $\|\mathbf{X}\|_2$, $\|\mathbf{X}\|_F$, $\|\mathbf{X}\|_\star$ and $\text{Tr}(\mathbf{X})$ represent the operator norm, the Frobenius norm, the nuclear norm and the trace of the matrix $\mathbf{X}$, respectively. $\|\mathbf{x}\|_{\ell_p}$ denotes the $\ell_p$-norm of the vector $\mathbf{x}$ and for matrices, $\|\mathbf{X}\|_{\ell_p} = \|\text{Vec}(\mathbf{X})\|_{\ell_p}$. For both vectors and matrices, $\|\cdot\|_0$ indicates the number of non-zero entries. The set of $n \times n$ positive definite matrices and positive semi-definite matrices are denoted by $\mathbb{S}_{++}^n$ and $\mathbb{S}_+^n$, respectively. The letters $\mathbf{g}$ and $\mathbf{G}$ are reserved for a Gaussian random vector and matrix with i.i.d. standard normal entries. The letter $\mathbf{H}$ is reserved for a random Gaussian *Wigner* matrix, that is a *symmetric* matrix whose upper-diagonal entries drawn independently from $\mathcal{N}(0,1)$ whose its diagonals entries are drawn independently from $\mathcal{N}(0,2)$. Finally, the letter $\mathbf{I}$ is reserved for the identity matrix. For a random vector $\mathbf{a}$, $\mathbb{E}[\mathbf{a}]$ and $\text{Cov}[\mathbf{a}]$ represent the expected value and the covariance matrix of $\mathbf{a}$.

## 2.2 Problem Setup

We consider the problem of recovering the unknown vector $\mathbf{x}_0 \in \mathcal{S} \subseteq \mathbb{R}^n$ from $m$ observations of the form $y_i = \mathbf{a}_i^\mathsf{T} \mathbf{x}_0$, $i = 1, \ldots, m$. Here, the *known* measurement vectors $\mathbf{a}_i \in \mathbb{R}^n$'s are drawn independently and identically from a random distribution. These observations can be reformulated as

$$\mathbf{y} = \mathbf{A}\mathbf{x}_0 \,, \tag{2}$$

where $\mathbf{y} = [y_1, \ldots, y_m]^\mathsf{T} \in \mathbb{R}^m$ and $\mathbf{A} = [\mathbf{a}_1, \ldots, \mathbf{a}_m]^\mathsf{T} \in \mathbb{R}^{m \times n}$. We focus on the high-dimensional setting where both $n$ and $m$ grow large. We use the notation $m = \theta(n)$, to fix the rate at which $m$ grows compared to $n$. Of special interest is the underdetermined case where the number of measurement is smaller than the ambient dimension. In this case, the problem of signal reconstruction is generally ill-posed unless some prior information is available regarding the structure of $\mathbf{x}_0$. Some popular cases of structures include, *sparse* vectors, *low-rank* matrices, and simultaneously-structured matrices.

**Convex estimator:** To recover the structured vector $\mathbf{x}_0$, we minimize a convex function $f : \mathbb{R}^n \to \mathbb{R}$ that enforces this structure. We do this minimization for all feasible points $\mathbf{x} \in \mathcal{S}$, that satisfy $\mathbf{y} = \mathbf{A}\mathbf{x}$. We formally define such estimators as follows,

**Definition 1.** *Let $\mathbf{x}_0 \in \mathcal{S}$ where $\mathcal{S} \subseteq \mathbb{R}^n$ is a convex set. For a convex function $f : \mathbb{R}^n \to \mathbb{R}$ and a measurement matrix $\mathbf{A} \in \mathbb{R}^{m \times n}$, we define the convex estimator $\mathcal{E}\{\mathbf{x}_0, \mathbf{A}, \mathcal{S}, f(\cdot)\}$ as following,*

$$\hat{\mathbf{x}} = \arg \min_{\substack{\mathbf{x} \in \mathcal{S} \\ \mathbf{A}\mathbf{x} = \mathbf{A}\mathbf{x}_0}} f(\mathbf{x}) \,. \tag{3}$$

*We say $\mathcal{E}\{\mathbf{x}_0, \mathbf{A}, \mathcal{S}, f(\cdot)\}$ has perfect recovery iff $\hat{\mathbf{x}} = \mathbf{x}_0$.*

Note that we are given the observation vector $\mathbf{y} = \mathbf{A}\mathbf{x}_0$ in the constraint of (3). We aim to characterize the perfect recovery criteria for this estimator. Given a structured vector $\mathbf{x}_0$, the perfect recovery of an estimator $\mathcal{E}\{\mathbf{x}_0, \mathbf{A}, \mathcal{S}, f(\cdot)\}$ depends on three factors; the number of observations $m$ compared to the dimension of the ambient space $n$, properties of the measurement vectors $\{\mathbf{a}_i\}_{i=1}^m$, and the penalty function, $f(\cdot)$. We briefly explain each factor, below.

**The rate function $\theta(\cdot)$:** We work in the high dimensional regime where both $n$ and $m$ grow to infinity with a fixed rate $m = \theta(n)$. Finding the minimum number of measurements to recover $\mathbf{x}_0$ via (3), translates to finding the *smallest rate function $\theta^\star(\cdot)$*, for which our estimator has perfect recovery. This optimal rate function depends on the problem settings and varies in different problems. For instance, in order to recover a rank-$r$ matrix in $\mathbb{S}_+^n$, we will need the measurements to be of order $m = \mathcal{O}(n)$, while in the case of $k$-sparse matrices, the measurements will be of order $m = \mathcal{O}(k \log(n^2/k))$, where in many applications $k$ is a fraction of $n^2$.

**The penalty function:** We use a convex function $f(\cdot)$ that promotes the particular structure of $\mathbf{x}_0$. Exploiting a convex penalty for the recovery of structured signals has been studied extensively [9, 1, 28, 14, 4, 29]. Chandrasekaran et. al. [9] introduced the concept of the atomic norm, which is a convex surrogate defined based on a set of (so-called) "atoms". For instance, the corresponding atomic norm for sparse recovery is the $\ell_1$-norm and for low-rank matrix recovery the nuclear norm. Another interesting scenario is when the underlying parameter $\mathbf{x}_0$ simultaneously exhibits multiple structures such as being low-rank and sparse. For simultaneously structured signals building the set of atoms is often intractable. Therefore, it has been proposed [21, 10] to use a weighted sum of corresponding atomic norms for each structure as the penalty.

**The measurement vectors:** We consider a random ensemble, where the vectors $\{\mathbf{a}_i\}_{i=1}^m$ are drawn *independently and identically* from a random distribution. Later in Section 2.3, we formally present the required assumptions on this distribution. It has been observed that the estimator (3) exhibits a *phase transition* phenomenon, i.e., there exist a phase transition rate $\theta^\star(n)$, such that when $m > \theta^\star(n)$ the optimization program (3) successfully recover $\mathbf{x}_0$ with high probability, otherwise,

when $m < \theta^\star(n)$ it fails with high probability [1, 9]. The question is that *how is this phase transition is related to the properties of the measurement vectors $\mathbf{a}_i$'s?*

**Universality in learning:** Directly calculating the precise phase transition behavior of the estimator $\mathcal{E}(\mathbf{x}_0, \mathbf{A}, \mathcal{S}, f(\cdot))$, for a general random distribution on the measurement vectors is very challenging. Recently, as an extension of Gaussian comparison lemmas due to Gordon [16, 17] and earlier work in [27, 28, 9, 1], a new framework, known as CGMT [29, 30], has been developed which made this analysis possible when the measurement vectors $\{\mathbf{a}_i\}_{i=1}^m$, are independently drawn from the Gaussian distribution, $\mathcal{N}(0, \mathbf{I}_n)$. Another parallel work that makes this analysis possible under the same conditions is known as AMP [14]. However, the Gaussian assumption is critical in the analysis through these frameworks, which restricts us from investigating a vast variety of practical problems. As our main result, we show that, for a broad class of distributions, the phase transition of $\mathcal{E}(\mathbf{x}_0, \mathbf{A}, \mathcal{S}, f(\cdot))$ depends only on the first two statistics of the distribution on the measurement vectors $\{\mathbf{a}_i\}_{i=1}^m$. As a result, the phase transition of the estimator remains unchanged when we replace the measurement vectors with the ones drawn from a Gaussian distribution with the same mean vector and covariance matrix. As the phase transition is the same as the one with Gaussian measurements, we can use the CGMT framework to analyze the latter and get the desired result.

**Equivalent Gaussian Problem:** Let $\mu := \mathbb{E}[\mathbf{a}_i]$ and $\mathbf{\Sigma} := \mathrm{Cov}[\mathbf{a}_i]$ for $i = 1, 2, \ldots, m$, and consider the following problem:

1. We are given $m$ observations of the form $\tilde{y}_i = \mathbf{g}_i^\mathsf{T} \mathbf{x}_0$ and the measurement vectors $\{\mathbf{g}_i\}_{i=1}^m$.

2. The rows of the measurement matrix $\mathbf{G} = [\mathbf{g}_1, \ldots, \mathbf{g}_m]^\mathsf{T} \in \mathbb{R}^{m \times n}$ are independently drawn from the multivariate Gaussian distribution $\mathcal{N}(\mu, \mathbf{\Sigma})$.

3. We use the estimator $\mathcal{E}(\mathbf{x}_0, \mathbf{G}, \mathcal{S}, f(\cdot))$, as in Definition 1, to recover $\mathbf{x}_0$.

In Theorem 1, we show that under certain conditions, the two estimators $\mathcal{E}(\mathbf{x}_0, \mathbf{A}, \mathcal{S}, f(\cdot))$ and $\mathcal{E}(\mathbf{x}_0, \mathbf{G}, \mathcal{S}, f(\cdot))$ asymptotically exhibit the same phase transition behavior. Before stating our main result in Section 3, we discuss the assumptions needed for our universality to hold.

## 2.3 Assumptions

We show universality for a wide range of distributions on the measurement vector as well as a broad class of convex penalties. Here, we give the conditions needed for the measurement matrix,

**Assumption 1.** *[The Measurement Vectors] We say the measurement matrix $\mathbf{A} = [\mathbf{a}_1, \ldots, \mathbf{a}_m]^\mathsf{T} \in \mathbb{R}^{m \times n}$ satisfies Assumption 1 with parameters $\mu \in \mathbb{R}^n$ and $\mathbf{\Sigma} \in \mathbb{R}^{n \times n}$, if the followings hold true.*

1. *[Sub-Exponential Tails] The vectors $\mathbf{a}_i$'s are independently drawn from a random sub-exponential distribution, with mean $\mu$ and covariance $\mathbf{\Sigma} \succ 0$.*

2. *[Bounded Mean] For some constants $c_1, \tau_1 > 0$, we have $\frac{\|\mu\|_2^2}{\mathbb{E}[\|\mathbf{a}_i - \mu\|^2]} \leq c_1 \cdot n^{-\tau_1}$, for all $i$.*

3. *[Bounded Power] For some constants $c_2, \tau_2 > 0$, we have $\frac{Var(\|\mathbf{a}_i\|^2)}{\mathbb{E}^2[\|\mathbf{a}_i - \mu\|^2]} \leq c_2 \cdot n^{-\tau_2}$ for all $i$.*

Assumption 1 summarizes the technical conditions that are essential in the proof of our main theorem. The first assumption on the tail of the distribution enables us to exploit concentration inequalities for sub-exponential distributions. We allow the vector $\mathbf{a}_i$ to have a non-zero mean in Assumption 1.2. Yet we require the power of its mean to be small compared to the power of the random part of the vector. Intuitively, one would like the measurement vectors to sample diversely from all the directions in $\mathbb{R}^n$, and not be biased towards a specific direction. Finally, Assumption 1.3 is meant to control the dependencies among the entries of $\mathbf{a}_i$ and is used to prove concentration of $\frac{1}{n} \mathbf{a}_i^\mathsf{T} \mathbf{M} \mathbf{a}_i$ around its mean, for a matrix $\mathbf{M}$ with bounded operator norm. For instance, for a Gaussian vector $\mathbf{g} \sim \mathcal{N}(\mathbf{0}, \mathbf{I})$, we have $Var[\|\mathbf{g}\|^2] = 2n$ and $\mathbb{E}^2[\|\mathbf{g}\|^2] = n^2$. So Assumption 1.3 is satisfied with $c_2 = 2$ and $\tau_2 = 1$. We will examine these assumptions for the applications discussed in Section 4.

In addition, we need to enforce a few conditions on the penalty function $f(\cdot)$ as follows,

**Assumption 2.** *[The Penalty Function] We say the funtion $f(\cdot)$ satisfies Assumption 2, if the following holds true.*

1. *[Separablity] $f(\cdot)$ is continuous, convex and separable, where $f(\mathbf{x}) = \sum_{i=1}^n f_i(x_i)$.*

2. *[Smoothness] The functions $\{f_i(\cdot)\}$ are three times differentiable everywhere, except for a finite number of points.*

3. *[Bounded Third Derivative] For any $C > 0$, there exists a constant $c_f > 0$, such that for all $i$, we have $|\frac{\partial^3 f_i(x)}{\partial x^3}| \le c_f$, for all smooth points in the domain of $f_i(\cdot)$ such that $|x| < C$.*

As observed in the Assumption 2.1, we only consider the special (yet popular) case of separable penalty functions. Common choices include $\|\mathbf{x}\|_{\ell_1}$ and $\|\mathbf{x}\|_{\ell_2}^2$ for vectors, and $\|\mathbf{X}\|_{\ell_1}$, $\|\mathbf{X}\|_F$ and $\text{Tr}(\mathbf{X})$ (which is equivalent to the nuclear norm of $\mathbf{X}$ when $\mathbf{X} \in \mathbb{S}_+$) for matrices. We can also apply our theorem for $\ell_p$-norm. This is due to the fact that replacing $\|\cdot\|_{\ell_p}$ with $\|\cdot\|_{\ell_p}^p$ does not change our estimate, and the latter is a separable function.

## 3  Main Result

In this section, we state our main theorem which shows that the performance of the convex estimator $\mathcal{E}(\mathbf{x}_0, \mathbf{A}, \mathcal{S}, f(\cdot))$, is independent of the distribution of the measurement vectors. So we can replace them with the Gaussian random vectors with the same mean and covariance. Next, using CGMT framework [29, 30], we analyze the phase transition in the case with Gaussian measurements, in Corollary 1. Later, we will apply this result to some well-known problems in Section 4.

### 3.1  Universality Theorem

**Theorem 1.** *[non-Gaussian=Gaussian] Consider the problem of recovering $\mathbf{x}_0 \in \mathcal{S} \subseteq \mathbb{R}^n$ from the measurements $\mathbf{y} = \mathbf{A}\mathbf{x}_0 \in \mathbb{R}^m$, using a convex penalty function $f(\cdot)$ in the estimator $\mathcal{E}\{\mathbf{x}_0, \mathbf{A}, \mathcal{S}, f(\cdot)\}$ in (3). Assume $\mathcal{S}$ is a convex set and $m$ and $n$ are growing to infinity at a fixed rate $m = \theta(n)$. Also assume that*

1. *$f : \mathbb{R}^n \to \mathbb{R}$ is a convex function that satisfies Assumption 2.*

2. *The measurement matrix $\mathbf{A} = [\mathbf{a}_1, \dots, \mathbf{a}_m]^\mathsf{T}$ satisfies Assumption 1, with $\mu := \mathbb{E}[\mathbf{a}_i]$ and $\boldsymbol{\Sigma} := \text{Cov}[\mathbf{a}_i]$ for all $i = 1, \dots, m$.*

3. *$\mathbf{G} = [\mathbf{g}_1, \dots, \mathbf{g}_m]^\mathsf{T} \in \mathbb{R}^{m \times n}$ is a random Gaussian matrix with independent rows drawn from Gaussian distribution $\mathcal{N}(\mu, \boldsymbol{\Sigma})$.*

*Then the estimator $\mathcal{E}\{\mathbf{x}_0, \mathbf{A}, \mathcal{S}, f(\cdot)\}$ (introduced in Definition 1) succeeds in recovering $\mathbf{x}_0$ with probability approaching one (as $m$ and $n$ grow large), if and only if the estimator $\mathcal{E}\{\mathbf{x}_0, \mathbf{G}, \mathcal{S}, f(\cdot)\}$ succeeds with probability approaching one.*

Theorem 1 shows that only the mean and covariance of the measurement vectors $\mathbf{a}_i$ affect the required number of measurements for perfect recovery in (3). Although Theorem 1 holds for $n$ and $m$ growing to infinity, the result of our numerical simulations in Section 3.2, indicates the validity of universality for values of $m$ and $n$ ranging in the order of hundreds.

#### 3.1.1  Analysis of the Gaussian Estimator

Theorem 1 shows the equivalence of the convex estimator $\mathcal{E}\{\mathbf{x}_0, \mathbf{A}, \mathcal{S}, f(\cdot)\}$ and the Gaussian estimator $\mathcal{E}\{\mathbf{x}_0, \mathbf{G}, \mathcal{S}, f(\cdot)\}$. We can utilize the CGMT framework to analyze the perfect recovery conditions for $\mathcal{E}\{\mathbf{x}_0, \mathbf{G}, \mathcal{S}, f(\cdot)\}$. Before doing so, we need the definition of the *descent cone*,

**Definition 2.** *[Descent Cone] The descent cone of a convex function $f(\cdot)$ at point $\mathbf{x}_0$ is defined as*

$$\mathcal{D}_f(\mathbf{x}_0) = Cone\left(\{\mathbf{y} \,:\, f(\mathbf{y}) \le f(\mathbf{x}_0)\}\right) , \tag{4}$$

*which is a convex cone. Here, $Cone(\mathcal{S})$ denotes the conic-hull of the set $\mathcal{S}$.*

**Corollary 1.** *Consider the problem of recovering the vector $\mathbf{x}_0 \in \mathcal{S}$, given the observations $\mathbf{y} = \mathbf{G}\mathbf{x}_0 \in \mathbb{R}^m$, via the estimator $\mathcal{E}\{\mathbf{x}_0, \mathbf{G}, \mathcal{S}, f(\cdot)\}$ introduced earlier. Assume that the rows of $\mathbf{G}$ are independent Gaussian random vectors with mean $\mu$ and covariance $\boldsymbol{\Sigma} = \mathbf{M}\mathbf{M}^\mathsf{T}$. Let $\delta := m/n$ and the set $\mathcal{S}$ and the penalty function $f(\cdot)$ be convex. $\mathcal{E}\{\mathbf{x}_0, \mathbf{G}, \mathcal{S}, f(\cdot)\}$ succeed in recovering $\mathbf{x}_0$ with probability approaching one (as $m$ and $n$ grow to infinity), if and only if*

$$\sqrt{\delta} > \sqrt{\delta^\star} = \mathbb{E}\left[ \max_{\substack{\mathbf{w} \in (\mathcal{S} - \mathbf{x}_0) \cap D_f(\mathbf{x}_0) \\ \frac{1}{\sqrt{n}}\mathbf{M}^\mathsf{T}\mathbf{w} \in S_{n-1}}} \frac{\mathbf{w}^\mathsf{T}\mathbf{g}}{n\sqrt{1 + \frac{1}{n}(\mathbf{w}^\mathsf{T}\mu)^2}} \right] \tag{5}$$

*where $S_{n-1}$ is the $n$-dimensional unit sphere, and the expected value is over the Gaussian vector $\mathbf{g} \sim \mathcal{N}(\mathbf{0}, \boldsymbol{\Sigma})$.*

**["Pseudo Gaussian Width"]** When $\mu = 0$ and $\Sigma = \mathbf{I}$, the expected value in (5) resembles the definition of the *Gaussian width* [25]. It has been shown that when the measurements are i.i.d. Gaussian, the square of the Gaussian width indicates the phase transition for linear inverse problems [9, 1, 28]. The Gaussian width has been computed for several interesting examples, such as sparse recovery, and low-rank matrix recovery. Using our universality result in Theorem 1, we can state that the square of the Gaussian width indicates the phase transition in the non-Gaussian setting as well.

### 3.2 Numerical Results

To validate the result of Theorem 1, we performed numerical simulations under various distributions for the measurement vectors. For our simulations in Figure 1, we use the estimator $\mathcal{E}\{\mathbf{x}_0, \mathbf{A}, \mathbb{R}^n, \|\cdot\|_{\ell_1}\}$ to recover a $k$-sparse signal $\mathbf{x}_0$ under three random ensembles for the measurement vectors $\{\mathbf{a}_i\}_{i=1}^m$. In each of the three plots, we computed the norm of the estimation error $\mathcal{E}\{\mathbf{x}_0, \mathbf{A}, \mathbb{R}^n, \|\cdot\|_{\ell_1}\}$, for different over sampling ratios $\delta = m/n$ and multiple sparsity factors $s = k/n$. We generated the measurement vectors $\{\mathbf{a}_i\}_{i=1}^m$ for each figure, as follows,

- For each trial, we generate a random matrix $\mathbf{M} \in \mathbb{R}^{n \times n}$, with i.i.d. standard Gaussian random variables. $\Sigma = \mathbf{M}\mathbf{M}^{\mathsf{T}}$ will play the role of the covariance matrix of the measurement vectors.
- For Figure 1a, $\{\mathbf{a}_i\}_{i=1}^m$ are drawn independently from the Gaussian distribution $\mathcal{N}(\mathbf{0}, \Sigma)$.
- For the measurement vectors of the Figure 1b, we first generate i.i.d centered bernouli vectors Ber(.8), and multiply each vector by $\mathbf{M}$.
- For the measurement vectors of the Figure 1c, we first generate i.i.d centered $\chi_1$ vectors, and multiply each vector by $\mathbf{M}$.

The blue line in the figures shows the theoretical phase transition derived as a result of Corollary 1. It can be observed that the phase transition for all the three random schemes is the same, as predicted by Theorem 1. It also matches the theoretical phase transition derived from Corollary 1.

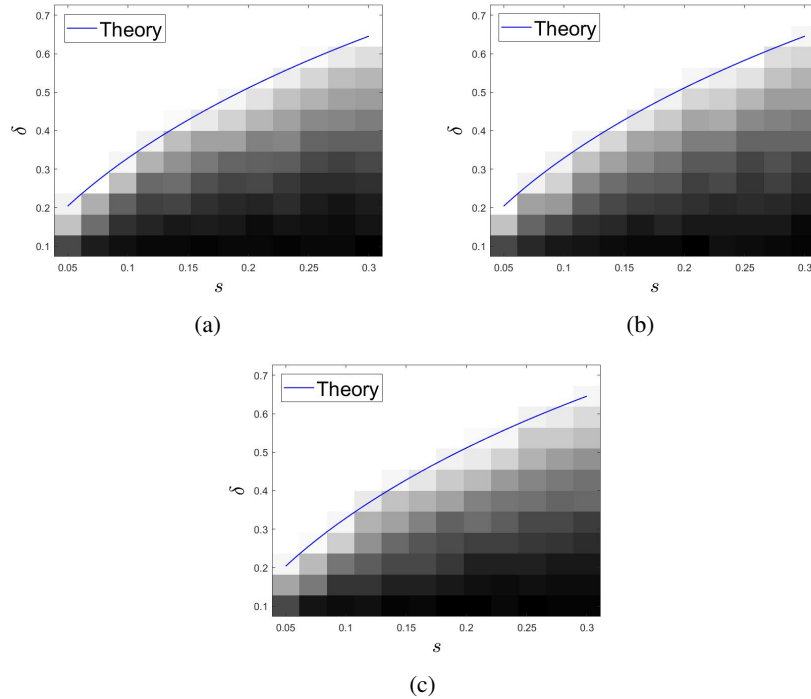

Figure 1: Phase transition regimes for the estimator $\mathcal{E}\{\mathbf{x}_0, \mathbf{A}, \mathbb{R}^n, \|\cdot\|_{\ell_1}\}$, in terms of the oversampling ratio $\delta = \frac{m}{n}$ and $s = \frac{\|\mathbf{x}_0\|_0}{n}$, for the cases of (a) Gaussian measurements and (b) Bernoulli measurements and (c) $\chi^2$ measurements. The blue lines indicate the theoretical estimate for the phase transition derived from Corollary 1. In the simulations we used vectors of size $n = 256$. The data is averaged over 10 independent realization of the measurements.

Next, to illustrate the applicability and the implications of the results, we present some examples where our universality theorem can be applied.

# 4 Applications: Quadratic Measurements

In this section we consider the problem of recovering a matrix from (so-called) *quadratic measurements*. The goal is to reconstruct a symmetric matrix $\mathbf{X}_0 \in \mathbb{R}^{n \times n}$ in a convex set $\mathcal{S}$, given $m$ measurements of the form,

$$y_i = \mathbf{a}_i^\mathsf{T} \mathbf{X}_0 \mathbf{a}_i = \mathrm{Tr}\left(\mathbf{X}_0 \cdot (\mathbf{a}_i \mathbf{a}_i^\mathsf{T})\right), \quad i = 1, \dots, m. \tag{6}$$

Depending on the application, the matrix $\mathbf{X}_0$ may exhibit various structures. Similar to (3), we use the convex penalty function $f : \mathbb{R}^{n \times n} \to n$, to enforce this structure via the following convex estimator,

$$\hat{\mathbf{X}} = \arg\min_{\mathbf{X} \in \mathcal{S}} \; f(\mathbf{X})$$
$$\text{subject to:} \quad \mathbf{a}_i^\mathsf{T} \mathbf{X} \mathbf{a}_i = \mathbf{a}_i^\mathsf{T} \mathbf{X}_0 \mathbf{a}_i, \quad i = 1, \dots, m. \tag{7}$$

Note that the measurements in (6) are linear with respect to the matrix $\mathbf{X}_0$, yet quadratic with respect to the measurement vectors $\mathbf{a}_i$. We can define $\tilde{\mathbf{x}}_0 := \mathrm{Vec}(\mathbf{X}_0) \in \mathbb{R}^{n^2}$ and $\tilde{\mathbf{a}}_i := \mathrm{Vec}(\mathbf{a}_i \mathbf{a}_i^\mathsf{T}) \in \mathbb{R}^{n^2}$, such that the measurements take the familiar form, $y_i = \tilde{\mathbf{a}}_i^\mathsf{T} \tilde{\mathbf{x}}_0$. In order to apply the result of Theorem 1, one should check if the vectors $\{\tilde{\mathbf{a}}_i\}_{i=1}^m$ satisfy Assumption 1.

It can be shown that if the vectors $\{\mathbf{a}_i\}_{i=1}^m$ satisfy the following conditions, then Assumption 1 holds true for $\{\tilde{\mathbf{a}}_i = \mathrm{Vec}(\mathbf{a}_i \mathbf{a}_i^\mathsf{T})\}_{i=1}^m$.

**Assumption 3.** *We say vectors $\{\mathbf{a}_i\}_{i=1}^m$ satisfy Assumption 3, if*

1. *$\mathbf{a}_i$'s are drawn independently from a sub-Gaussian distribution.*

2. *For each $i$, the entries of $\mathbf{a}_i$ are independent, zero-mean and unit-variance.*

In particular, this assumption is valid when $\{\mathbf{a}_i\}$'s have i.i.d. standard normal entries. Therefore, when Assumption 3 holds, we can apply Theorem 1 to show that the required number of measurements for perfect recovery in (7) is equal to the required number of measurements for the success of the following estimator,

$$\hat{\mathbf{X}} = \arg\min_{\mathbf{X} \in \mathcal{S}} \; f(\mathbf{X})$$
$$\text{subject to:} \quad \mathrm{Tr}\left((\mathbf{H}_i + \mathbf{I})\mathbf{X}\right) = \mathrm{Tr}\left((\mathbf{H}_i + \mathbf{I})\mathbf{X}_0\right), \quad i = 1, \dots, m, \tag{8}$$

where $\mathbf{I}$ is the $n \times n$ identity matrix and $\mathbf{H}_i$'s are independent Gaussian Wigner matrices (defined in Section 2). Corollary 2 presents a formal statement.

**Corollary 2.** *Consider the problem of recovering the matrix $\mathbf{X}_0 \in \mathcal{S} \subseteq \mathbb{R}^{n \times n}$, from $m$ quadratic measurements of the form (6), using the estimator (7). Let $\mathcal{S}$ and $f(\cdot)$ be convex set and function satisying Assumption 2. Assume,*

- *The measurement vectors $\{\mathbf{a}_i\}_{i=1}^m$ satisfy Assumption 3, and,*

- *$\{\mathbf{H}_i \in \mathbb{R}^{n \times n}\}_{i=1}^m$ is a set of independent Gaussian Wigner matrices.*

*Then, as $m$ and $n$ grow to infinity at a fixed rate $m = \theta(n)$, the estimator (7) perfectly recovers $\mathbf{X}_0$ with probability approaching one if and only if the estimator (8) perfectly recovers $\mathbf{X}_0$ with probability approaching one.*

Therefore, in order to find the phase transition, it is sufficient to analyze the equivalent optimization (8) which is possible via the CGMT framework. Proceeding onward, we exploit the CGMT framework along with Corollary 1 to find the required number of measurements for the recovery of $\mathbf{X}_0$ in two specific applications.

## 4.1 Low-rank Matrix Recovery

Assume the unknown matrix $\mathbf{X}_0 \succeq \mathbf{0}$ has rank $r$, where $r$ is a constant ( i.e., $r$ does not grow with problem dimensions $n, m$.) Such matrices appear in many applications such as traffic data monitoring, array signal processing and phase retrieval. The nuclear norm, $|| \cdot ||_\star$, is often used as the convex surrogate for low-rank matrix recovery [24]. Hence, we are interested in analyzing the optimization

(7), with the choice of $f(\mathbf{X}) = \|\mathbf{X}\|_\star$, where the optimization is over the set of PSD matrices. Note that $\mathrm{Tr}(\cdot) = \|\cdot\|_\star$ within this set, which satisfies Assumption 2.

According to Corollary 2, the perfect recovery in (7) is equivalent to perfect recovery in (8), where the same choice of $f(\mathbf{X}) = \mathrm{Tr}(\mathbf{X})$. The analysis of the later through CGMT yields the following corollary.

**Corollary 3.** *Consider the optimization program (7), where the matrix $\mathbf{X}_0 \succeq 0$ has rank $r$, $f(\mathbf{X}) = \mathrm{Tr}(\mathbf{X})$, the set $\mathcal{S}$ is the PSD cone and the measurement vectors $\{\mathbf{a}_i\}_{i=1}^m$ satisfy Assumption 3. Assume $m, n \to \infty$ at the proportional rate $\delta := \frac{m}{n} \in (0, +\infty)$. The estimator perfectly recovers $\mathbf{X}_0$ if $\delta > 3r$.*

Corollary 3 indicates that $3rn$ measurements is needed to perfectly recover a rank-$r$ PSD matrix $\mathbf{X}_0$, from quadratic measurements. Although, the error of estimation gets extremely small, much before the threshold $m = 3nr$. To the extent of our knowledge, this is the first work that precisely computes the phase transition of low-rank matrix recovery from quadratic measurements. Figure2 depicts the result of numerical simulations. For different values of $r$ and $\delta$, the Frobenius norm of the error of the estimators (7) and (8) has been computed, which shows the same phase transition in both cases.

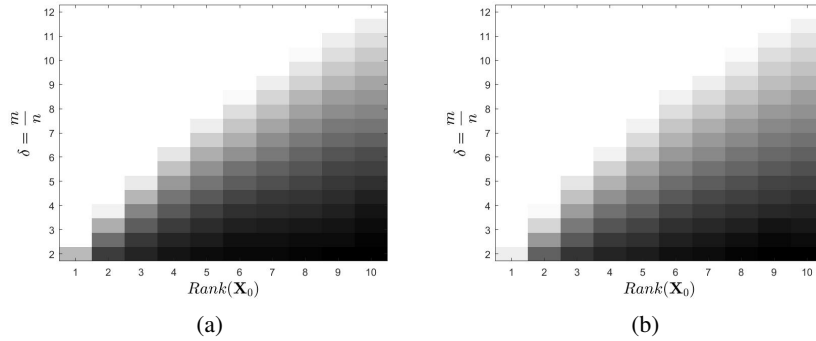

(a)           (b)

Figure 2: Phase transition regimes for both estimators 7 and (8), with $f(\mathbf{X}) = \mathrm{Tr}(\mathbf{X})$, in terms of the oversampling ratio $\delta = \frac{m}{n}$ and $r = \mathrm{Rank}(\mathbf{X}_0)$, for the cases of (a) estimator (7) with quadratic measurements and (b) estimator (8) with Gaussian measurements. In the simulations we used matrices of size $n = 40$. The data is averaged over 20 independent realization of the measurements.

### 4.1.1 Phase Transition of PhaseLift in Phase Retrieval

An important application for the result of Corollary 3, is when the underlying matrix $\mathbf{X}_0$ is of rank 1. This appears in the problem of phase retrieval, where $\mathbf{X}_0 = \mathbf{x}_0 \mathbf{x}_0^T$ is the lifted version of the signal. The optimization program (7) with $f(\mathbf{X}) = \mathrm{Tr}(\mathbf{X})$ in this case, is known as PhaseLift [7]. Corollary 3 states that the phase transition of the PhaseLift algorithm happens at $\delta^\star = 3$, i.e., $m > 3n$ measurements is needed for the perfect signal reconstruction in PhaseLift. We should emphasize the significance of this result as establishing the exact phase transition of the PhaseLift algorithm was long an open problem.

### 4.2 Sparse Matrix Recovery

Let $\mathbf{X}_0 \succeq 0$ represent the covariance matrix of a set of random variables. In certain applications, the covariance matrix has many near-zero entries as the correlations are small for many pairs of random variables. Such matrices arise in applications in spectrum estimation, biology and finance [15, 11]. We are interested in analyzing estimator (7), where $f(\mathbf{X}) = \|\mathbf{X}\|_{\ell_1}$ promotes the sparsity in the optimization. As $\|\cdot\|_{\ell_1}$ satisfies Assumption 2, applying the result of Corollary 2, the perfect recovery in (7) is equivalent to the perfect recovery in the estimator (8), with the same penalty function. Analyzing the optimization (8) via CGMT leads to the following result:

**Corollary 4.** *Let $\delta := \frac{m}{n^2}$, $s := \frac{\|\mathbf{X}_0\|_0}{n^2}$. As $n \to \infty$, the optimization program (7), with $f(\mathbf{X}) = \|\mathbf{X}\|_{\ell_1}$ can successfully recover the signal iff $\delta > \delta^\star$, where $\delta^\star$ is the unique solution to the following nonlinear equation,*

$$x \cdot Q^{-1}\left(\frac{2x-s}{2-2s}\right) = (1-s)\phi\left(Q^{-1}\left(\frac{2x-s}{2-2s}\right)\right), \tag{9}$$

| Model | Penalty function $f(\cdot)$ | No. of required measurements |
|---|---|---|
| $k$ sparse matrix | $\|\cdot\|_{\ell_1}$ | $n^2\delta^\star$ defined in (9) |
| Rank-$r$ PSD matrix | $\mathrm{Tr}(\cdot)$ | $3nr$ |
| S&L $(k,r)$ matrix | $\mathrm{Tr}(\cdot) + \lambda\|\cdot\|_1$ | $\mathcal{O}(\min(k^2, rn))$ |

Table 1: Summary of the parameters that are discussed in this section. The last row is for a $n \times n$ rank-$r$ matrix whose smallest sub-matrix with non-zero entries is $k$ by $k$. The third column shows the number of required quadratic measurements for perfect recovery.

where $\phi(x) = \exp(-x^2/2)/\sqrt{2\pi}$ and $Q^{-1}(\cdot)$ is inverse of the Q-function.

Figure 3b compares the empirical result with the theoretical phase transition derived from Corollary 4 Each plot shows the norm of the error with respect to the sparsity of the matrix $\mathbf{X}_0$ and the ratio $\delta = \frac{m}{n^2}$. A comparison between the two plots indicates that the phase transitions of the two estimators (7) and (8) with $f(\mathbf{X}) = \|\mathbf{X}\|_{\ell_1}$ match.

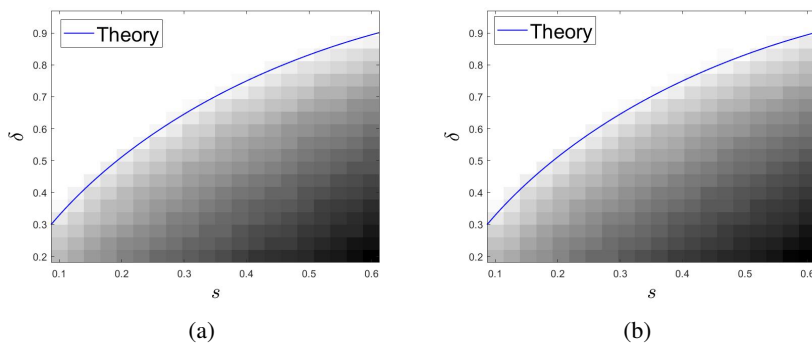

(a)                        (b)

Figure 3: Phase transition regimes for both estimators (7) and (8), with $f(\mathbf{X}) = \|\mathbf{X}\|_{\ell_1}$, in terms of the oversampling ratio $\delta = \frac{m}{n}$ and $s = \frac{\|\mathbf{X}_0\|_0}{n^2}$, for the cases of (a) estimator (7) with quadratic measurements and (b) estimator (8) with Gaussian measurements. The blue lines indicate the theoretical estimate for the phase transition derived from equation (9). In the simulations we used matrices of size $n = 40$. The data is averaged over 20 independent realization of the measurements.

## 4.3 Conclusion

We have investigated an estimation problem under linear observations. We aimed to characterize the minimum number of observations that are needed for perfect recovery of the unknown model. Our main result indicated that this phase transition, only depends on the first two statistics of the measurement vector. Therefore, it remains unchanged as we replace these vectors with the Gaussian one, with the same mean vector and covariance matrix. The later can be analyzed through existing frameworks such as CGMT. As one of the applications of this universality, we investigated the case of matrix recovery via the so called quadratic measurements, and derived the minimum number of observations required for the recovery of a structured matrix. Due to the space constraint, we moved the discussions regarding the case of simultaneously structured matrices to the appendix. Table 1, summarizes these results for the cases of three structures.

## Footnotes

[2]The reader should pardon the abuse of notation as the measurement vectors are now $\text{vec}(\mathbf{a}_i\mathbf{a}_i^t)$.

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
