[Supplementary Material · Sup.pdf]

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

$\quad\quad\quad\quad\quad\quad$ (a) $\quad\quad\quad\quad\quad\quad\quad\quad\quad\quad\quad\quad\quad\quad$ (b)

Figure 3: Phase transition regimes for both estimators (7) and (8), with $f(\mathbf{X}) = \|\mathbf{X}\|_{\ell_1}$, in terms of the oversampling ratio $\delta = \frac{m}{n}$ and $s = \frac{\|\mathbf{X}_0\|_0}{n^2}$, for the cases of (a) estimator (7) with quadratic measurements and (b) estimator (8) with Gaussian measurements. The blue lines indicate the theoretical estimate for the phase transition derived from equation (9). In the simulations we used matrices of size $n = 40$. The data is averaged over 20 independent realization of the measurements.

## 4.3   Conclusion

We have investigated an estimation problem under linear observations. We aimed to characterize the minimum number of observations that are needed for perfect recovery of the unknown model. Our main result indicated that this phase transition, only depends on the first two statistics of the measurement vector. Therefore, it remains unchanged as we replace these vectors with the Gaussian one, with the same mean vector and covariance matrix. The later can be analyzed through existing frameworks such as CGMT. As one of the applications of this universality, we investigated the case of matrix recovery via the so called quadratic measurements, and derived the minimum number of observations required for the recovery of a structured matrix. Due to the space constraint, we moved the discussions regarding the case of simultaneously structured matrices to the appendix. Table 1, summarizes these results for the cases of three structures.

## Footnotes

[1]The reader should pardon the abuse of notation as the measurement vectors are now $\mathrm{vec}(\mathbf{a}_i \mathbf{a}_i^t)$.

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

# 5 Simultaneously Sparse and Low-rank Matrices

Another interesting example is where the unknown matrix $\mathbf{X}_0 \succeq 0$ is simultaneously sparse and low rank. To recover $\mathbf{X}_0$, we would like to simultaneously minimize the penalty functions $f^{(1)}(\mathbf{X}) = \|\mathbf{X}\|_{\ell_1}$ and $f^{(2)}(\mathbf{X}) = \|\mathbf{X}\|_\star$, for all feasible matrices $\mathbf{X} \in \mathcal{S}$ that align our measurements in (6). Here, each function $f^{(i)}(\cdot)$ enforces one of the structures on $\mathbf{X}$. So, a natural choice for the regularizer function in (7) would be $f(\mathbf{X}) = f^{(1)}(\mathbf{X}) + \lambda f^{(2)}(\mathbf{X})$, where $\lambda$ is a regularizing parameter. Oymak et al [21] studied phase transition for perfect recovery of simultaneously structured matrices. Their results are based on Gordon's comparison lemma which is only applicable to the cases of linear Gaussian measurements. We can use the result of Corollary 2 to extend their result to settings with quadratic measurements, as the phase transition regime is equivalent in both cases. Let $\mathbf{X}_0 \in \mathbb{R}^{n \times n}$ be a rank-$r$ PSD matrix. Also assume that the largest sub-matrix in $\mathbf{X}_0$ that contains all non-zero entries is $k$ by $k$. If we choose $f(\mathbf{X}) = \|\mathbf{X}\|_{\ell_1} + \lambda \text{Tr}(\mathbf{X})$, they show that $\mathcal{O}(\min(k^2, rn))$ measurements is required for perfect recovery.

# 6 Proofs

## 6.1 Proof of Theorem 1

Consider the following optimization

$$\Phi_1 = \min_{\mathbf{A}\mathbf{x}_0 = \mathbf{A}\mathbf{x}} f(\mathbf{x}) \,, \tag{10}$$

Without loss of generality, assume that $f(0) = 0$. We change the variable to $\mathbf{w} = \mathbf{x} - \mathbf{x}_0$, which gives the following

$$\Phi_1 = \min_{\mathbf{A}\mathbf{w}=0} f(\mathbf{w} + \mathbf{x}_0) \,, \tag{11}$$

This optimization has perfect recovery, iff $\hat{\mathbf{w}} = 0$, or equivalently iff $\Phi_1 = 0$. We would like to show that if $\Phi_1 = 0$ with probability converging to 1, then the same holds if we replace the measurements vectors $\mathbf{a}_i$, with another set of measurement vectors with the same mean and covariance. We rewrite this optimization in the form of this min-max optimization,

$$\Phi_1 = \sup_{\lambda>0} \min_{\mathbf{w}} \frac{\lambda}{2}\|\mathbf{A}\mathbf{w}\|^2 + f(\mathbf{w} + \mathbf{x}_0)$$

$$= \sup_{\lambda>0} \min_{\mu>0} \min_{\mathbf{w}} \frac{\lambda}{2}\|\mathbf{A}\mathbf{w}\|^2 + f(\mathbf{w} + \mathbf{x}_0) + \frac{1}{2\mu}\|\mathbf{w}\|^2$$

$$= \sup_{\lambda>0} \lambda \cdot \min_{\mu>0} \min_{\mathbf{w}} \frac{1}{2}\|\mathbf{A}\mathbf{w}\|^2 + \frac{1}{\lambda}f(\mathbf{w} + \mathbf{x}_0) + \frac{1}{2\lambda\mu}\|\mathbf{w}\|^2 \tag{12}$$

Informally, we first show that for fixed values of $\lambda$ and $\mu$, the values of last minimization remains unchanged as we change the random measurement vectors inside it (as $m$ and $n$ grow to infinity). Next, we use Lemma **??** (See [29] Section A.4 and B.5) to switch the min-max over $\mu$ and $\lambda$, with the limit over $m$ and $n$.

By fixing the values of $\lambda$ and $\mu$, from now on, we redefine the function $f(\cdot)$ to be $\frac{1}{\lambda}f(\mathbf{w} + \mathbf{x}_0) + \frac{1}{2\lambda\mu}\|\mathbf{w}\|^2$, which is strongly convex. Note that we would like the following assumptions holds for these two set of random measurement vectors.

**Assumption 1:** Assume $\mathbf{A} = [\mathbf{a}_1, \ldots, \mathbf{a}_m]^\mathsf{T} \in \mathbb{R}^{m \times n}$ and $\mathbf{B} = [\mathbf{b}_1, \ldots, \mathbf{b}_m]^\mathsf{T} \in \mathbb{R}^{m \times n}$ are two random matrices, such that

$$\mathbf{e} = \mathbb{E}\left[\mathbf{a}_i\right] = \mathbb{E}\left[\mathbf{b}_i\right] \quad \forall i$$

$$\boldsymbol{\Sigma} = \mathbb{E}\left[\mathbf{a}_i \mathbf{a}_i^\mathsf{T}\right] = \mathbb{E}\left[\mathbf{b}_i \mathbf{b}_i^\mathsf{T}\right] \quad \forall i$$

$$\lim_{n\to\infty} \frac{\|\mathbf{e}\|^2}{n^2} = 0, \tag{13}$$

Besides, there exists $\tau > 0$ such that for any matrix $\mathbf{M} \in \mathbb{R}^{n \times n}$ such that $\|\mathbf{M}\|_2 \leq \kappa$, there exists some $c$ that only depends on $\kappa$ that

$$\frac{1}{n^2}\operatorname{Var}\left(\mathbf{a}_i^\mathsf{T}\mathbf{M}\mathbf{a}_i\right) \leq c \cdot n^{-\tau} \quad \text{and,}$$

$$\frac{1}{n^2}\operatorname{Var}\left(\mathbf{b}_i^\mathsf{T}\mathbf{M}\mathbf{b}_i\right) \leq c \cdot n^{-\tau} \,. \tag{14}$$

Now we want to investigate equivalence of the following two optimizations. Let $\mathbf{A} = [\mathbf{a}_1, \ldots, \mathbf{a}_m]$ and $\mathbf{B} = [\mathbf{b}_1, \ldots, \mathbf{b}_m]$ be $m$ by $n$ measurement matrices and

$$\Phi_{\mathbf{B}} = \min_{\mathbf{w}} \frac{1}{2m} \sum_{i=1}^{m} \left(z_i - \mathbf{w}^\mathsf{T}\mathbf{a}_i\right)^2 + f\left(\mathbf{w} + \mathbf{x}_0\right) \,,$$

$$\Phi_{\mathbf{A}} = \min_{\mathbf{w}} \frac{1}{2m} \sum_{i=1}^{m} \left(z_i - \mathbf{w}^\mathsf{T}\mathbf{b}_i\right)^2 + f\left(\mathbf{w} + \mathbf{x}_0\right) \,. \tag{15}$$

**Theorem 2.** *Consider the optimizations in* (**??**). *If*

$$\lim_{n,m\to\infty} \left|\mathbb{E}\left[\Phi_{\mathbf{B}} - \Phi_{\mathbf{A}}\right]\right| = 0 \,, \tag{16}$$

*and if for constants $C$ and $\delta > 0$,*

$$Pr\left(\left|\Phi_{\mathbf{A}} - C\right| > \delta\right) \xrightarrow{P} 0 \,, \tag{17}$$

*as $n, m \to \infty$. Then,*

$$Pr\left(\left|\Phi_{\mathbf{B}} - C\right| > 3\delta\right) \xrightarrow{P} 0 \,, \tag{18}$$

*Proof.* We first define the function $g : \mathbb{R} \to \mathbb{R}$ as follows.

$$
g(x) = \begin{cases} 0 & \text{if} \quad |x| \le 1, \\ (|x| - 1)^2 & \text{if} \quad 1 < |x| \le 2, \\ 2 - (|x| - 3)^2 & \text{if} \quad 2 < |x| \le 3, \\ 2 & \text{if} \quad |x| > 3 . \end{cases} \tag{19}
$$

Note that $g(.)$ is continuously differentiable with its first derivative bounded by 2. Now,

$$
\begin{aligned}
\Pr\{|\Phi_{\mathbf{B}} - C| > 3\delta\} = \Pr\left\{ g\left(\frac{\Phi_{\mathbf{B}} - C}{\delta}\right) > 2 \right\} &\le \frac{1}{2}\mathbb{E}\left[ g\left(\frac{\Phi_{\mathbf{B}} - C}{\delta}\right) \right] \\
&\le \frac{1}{2}\mathbb{E}\left[ g\left(\frac{\Phi_{\mathbf{A}} - C}{\delta}\right) \right] + \frac{1}{2}\left| \mathbb{E}\left[ g\left(\frac{\Phi_{\mathbf{A}} - C}{\delta}\right) - g\left(\frac{\Phi_{\mathbf{B}} - C}{\delta}\right) \right] \right| \\
&\le \Pr\{|\Phi_{\mathbf{A}} - C| > \delta\} + \frac{1}{2}\left| \mathbb{E}\left[ g'(\zeta) \cdot \left(\frac{\Phi_{\mathbf{A}} - C}{\delta} - \frac{\Phi_{\mathbf{B}} - C}{\delta}\right) \right] \right| \\
&\le \Pr\{|\Phi_{\mathbf{A}} - C| > \delta\} + \frac{1}{\delta}\left| \mathbb{E}[\Phi_{\mathbf{A}} - \Phi_{\mathbf{B}}] \right| \xrightarrow{n,m\to\infty} 0
\end{aligned} \tag{20}
$$

$\square$

**Theorem 3.** *Consider the optimizations in (**??**). If* $\mathbf{A}$, $\mathbf{B}$ *and* $f(.)$ *satisfy Assumption 1 and 2, respectively, then*

$$
\lim_{n,m\to\infty} |\mathbb{E}[\Phi_{\mathbf{A}}] - \mathbb{E}[\Phi_{\mathbf{B}}]| \to 0 . \tag{21}
$$

*Proof.* For $k = 0, \ldots, m$, we define

$$
\Phi_k := \min_{\mathbf{w}} \frac{1}{2m}\sum_{i=1}^{k}\left(z_i - \mathbf{a}_i^{\mathsf{T}}\mathbf{w}\right)^2 + \frac{1}{2m}\sum_{i=k+1}^{m}\left(z_i - \mathbf{b}_i^{\mathsf{T}}\mathbf{w}\right)^2 + f(\mathbf{w} + \mathbf{x}_0) . \tag{22}
$$

We have

$$
|\mathbb{E}[\Phi_{\mathbf{A}} - \Phi_{\mathbf{B}}]| = |\mathbb{E}[\Phi_m - \Phi_0]| \le \sum_{k=1}^{m} |\mathbb{E}[\Phi_k - \Phi_{k-1}]| . \tag{23}
$$

Now it suffices to show that there exists a constant $c$, such that for any $k$,

$$
|\mathbb{E}[\Phi_k - \Phi_{k-1}]| \le c\, m^{-(1+\tau/2)} , \tag{24}
$$

for some positive constant $\tau$. Since, then combining (**??**) and (**??**) yields,

$$
|\mathbb{E}[\Phi_{\mathbf{A}} - \Phi_{\mathbf{B}}]| \le \sum_{k=1}^{m} |\mathbb{E}[\Phi_k - \Phi_{k-1}]| \le c\, m^{-\tau/2} \to 0 . \tag{25}
$$

Let

$$
\begin{aligned}
\mathbf{M}_k &= [\mathbf{a}_1, \ldots, \mathbf{a}_{k-1}, \mathbf{b}_{k+1}, \ldots, \mathbf{b}_m]^{\mathsf{T}} \in \mathbb{R}^{(m-1)\times n}, \quad \text{and,} \\
\mathbf{z}_k &= [z_1, \ldots, z_{k-1}, z_{k+1}, \ldots, z_m]^{\mathsf{T}} \in \mathbb{R}^{m-1} .
\end{aligned} \tag{26}
$$

This helps us rewrite $\Phi_k$ and $\Phi_{k-1}$ as

$$
\begin{aligned}
\Phi_k &= \min_{\mathbf{w}} \frac{1}{2m}\|\mathbf{z}_k - \mathbf{M}_k\mathbf{w}\|^2 + \frac{1}{2m}\left(z_k - \mathbf{a}_k^{\mathsf{T}}\mathbf{w}\right)^2 + f(\mathbf{w} + \mathbf{x}_0) , \\
\Phi_{k-1} &= \min_{\mathbf{w}} \frac{1}{2m}\|\mathbf{z}_k - \mathbf{M}_k\mathbf{w}\|^2 + \frac{1}{2m}\left(z_k - \mathbf{b}_k^{\mathsf{T}}\mathbf{w}\right)^2 + f(\mathbf{w} + \mathbf{x}_0) .
\end{aligned} \tag{27}
$$

As of this point, we fix $k$ and drop the subscript $k$ from $z_k$, $\mathbf{z}_k$, $\mathbf{M}_k$, $\mathbf{a}_k$ and $\mathbf{b}_k$ for simplicity. The expectation in (**??**) is over the randomness in $z$, $\mathbf{z}$, $\mathbf{M}$, $\mathbf{a}$ and $\mathbf{b}$, which can be written as

$$
|\mathbb{E}[\Phi_k - \Phi_{k-1}]| = \left| \mathbb{E}_{\{\mathbf{M},\mathbf{z}\}}\left[ \mathbb{E}_{\{z,\mathbf{a},\mathbf{b}\}}\left[ \Phi_k - \Phi_{k-1} \big| \{\mathbf{M}, \mathbf{z}\} \right] \right] \right| \le \mathbb{E}_{\{\mathbf{M},\mathbf{z}\}}\left[ \left| \mathbb{E}_{\{z,\mathbf{a},\mathbf{b}\} \big| \{\mathbf{M},\mathbf{z}\}}[\Phi_k - \Phi_{k-1}] \right| \right] . \tag{28}
$$

We first fix $\mathbf{M}$ and $z$, and bound the inner expectation in (??). Now let,

$$\phi(\mathbf{a}, z, \mathbf{w}) = \frac{1}{2m}\|\mathbf{z} - \mathbf{M}\mathbf{w}\|^2 + \frac{1}{2m}\left(z - \mathbf{a}^\mathsf{T}\mathbf{w}\right)^2 + f\left(\mathbf{w} + \mathbf{x}_0\right),$$

$$\Phi(\mathbf{a}, z) = \min_{\mathbf{w}}\ \phi(\mathbf{a}, \mathbf{w})\,,$$

$$\bar{\Phi} = \Phi(\mathbf{0}, 0),\quad \text{and,}\quad \bar{\mathbf{w}} = \arg\min\ \phi(\mathbf{0}, 0, \mathbf{w})\,. \tag{29}$$

With these new definitions, we have $\Phi_k = \Phi(\mathbf{a}, z)$ and $\Phi_{k-1} = \Phi(\mathbf{b}, z)$ and thus,

$$\left|\mathbb{E}_{\{z,\mathbf{a},\mathbf{b}\}}\left[\Phi_k - \Phi_{k-1}\right]\right| = \left|\mathbb{E}_{\{z,\mathbf{a},\mathbf{b}\}}\left[\Phi(\mathbf{a}, z) - \Phi(\mathbf{b}, z)\right]\right|$$

$$\leq \left|\mathbb{E}_{\{z,\mathbf{a}\}}\left[\Phi(\mathbf{a}, z) - \bar{\Phi} - \frac{\sigma^2 + \frac{\|\bar{\mathbf{w}}\|^2}{m}}{2m(1 + \mathbb{E}[\mathbf{b}^\mathsf{T}\boldsymbol{\Omega}\mathbf{b}])}\right]\right|$$

$$+ \left|\mathbb{E}_{\{z,\mathbf{b}\}}\left[\Phi(\mathbf{b}, z) - \bar{\Phi} - \frac{\sigma^2 + \frac{\|\bar{\mathbf{w}}\|^2}{m}}{2m(1 + \mathbb{E}[\mathbf{b}^\mathsf{T}\boldsymbol{\Omega}\mathbf{b}])}\right]\right| \tag{30}$$

So since $\mathbb{E}[\mathbf{b}^\mathsf{T}\boldsymbol{\Omega}\mathbf{b}] = \mathbb{E}[\mathbf{a}^\mathsf{T}\boldsymbol{\Omega}\mathbf{a}]$, it remains to show that for positive constants $c$ and $\tau$,

$$\left|\mathbb{E}_{\{z,\mathbf{a}\}}\left[\Phi(\mathbf{a}, z) - \bar{\Phi} - \frac{\sigma^2 + \frac{\|\bar{\mathbf{w}}\|^2}{m}}{2m(1 + \mathbb{E}[\mathbf{a}^\mathsf{T}\boldsymbol{\Omega}\mathbf{a}])}\right]\right| \leq c\, m^{-(1+\tau/2)}\,,\quad \text{and,}$$

$$\left|\mathbb{E}_{\{z,\mathbf{b}\}}\left[\Phi(\mathbf{b}, z) - \bar{\Phi} - \frac{\sigma^2 + \frac{\|\bar{\mathbf{w}}\|^2}{m}}{2m(1 + \mathbb{E}[\mathbf{b}^\mathsf{T}\boldsymbol{\Omega}\mathbf{b}])}\right]\right| \leq c\, m^{-(1+\tau/2)}\,. \tag{31}$$

We show the later, and the proof of the first is similar. Define $\mathbf{v} = \frac{\partial f(\bar{\mathbf{w}} + \mathbf{x}_0)}{\partial \mathbf{w}}$ and $\mathbf{V} = \frac{\partial^2 f(\bar{\mathbf{w}} + \mathbf{x}_0)}{\partial \mathbf{w}^2}$ and

$$\psi(\mathbf{b}, z, \mathbf{w}) = \frac{1}{2m}\|\mathbf{z} - \mathbf{M}\mathbf{w}\|^2 + \frac{1}{2m}\left(z - \mathbf{b}^\mathsf{T}\mathbf{w}\right)^2 + f\left(\bar{\mathbf{w}} + \mathbf{x}_0\right) + \mathbf{v}^\mathsf{T}(\mathbf{w} - \bar{\mathbf{w}}) + \frac{1}{2}(\mathbf{w} - \bar{\mathbf{w}})^\mathsf{T}\mathbf{V}(\mathbf{w} - \bar{\mathbf{w}})\,,$$

$$\Psi(\mathbf{b}, z) = \min_{\mathbf{w}}\ \psi(\mathbf{b}, z, \mathbf{w})\,,\quad \text{and,}\quad \tilde{\mathbf{w}} = \arg\min\ \psi(\mathbf{b}, z, \mathbf{w})\,. \tag{32}$$

Note that by writing the optimality conditions, it is easy to show that $\Psi(\mathbf{0}, 0) = \Phi(\mathbf{0}, 0) = \bar{\Phi}$. Thus,

$$\mathbb{E}_{\{z,\mathbf{b}\}}\left|\left[\Phi(\mathbf{b}, z) - \bar{\Phi} - \frac{\sigma^2 + \frac{\|\bar{\mathbf{w}}\|^2}{m}}{2m(1 + \mathbb{E}[\mathbf{b}^\mathsf{T}\boldsymbol{\Omega}\mathbf{b}])}\right]\right| \leq \mathbb{E}_{\{z,\mathbf{b}\}}\left[|\Phi(\mathbf{b}, z) - \Psi(\mathbf{b}, z)|\right]$$

$$+ \left|\mathbb{E}_{\{z,\mathbf{b}\}}\left[\Psi(\mathbf{b}, z) - \Psi(\mathbf{0}, 0) - \frac{\sigma^2 + \frac{\|\bar{\mathbf{w}}\|^2}{m}}{2m(1 + \mathbb{E}[\mathbf{b}^\mathsf{T}\boldsymbol{\Omega}\mathbf{b}])}\right]\right|\,. \tag{33}$$

So we have to bound the two terms on the right hand side of (??). We start with bounding $\mathbb{E}_{\{z,\mathbf{b}\}}\left[|\Phi(\mathbf{b}, z) - \Psi(\mathbf{b}, z)|\right]$. Note that for any $\mathbf{w}$ we have

$$|\psi(\mathbf{b}, z, \mathbf{w}) - \phi(\mathbf{b}, z, \mathbf{w})| \leq \frac{C_f}{m}\|\mathbf{w} - \bar{\mathbf{w}}\|_3^3\,. \tag{34}$$

Besides, due to strong convexity of $\bar{f}(.)$ we have,

$$|\psi(\mathbf{b}, z, \mathbf{w}) - \Psi(\mathbf{b}, z)| \geq \frac{\epsilon}{m}\|\mathbf{w} - \tilde{\mathbf{w}}\|_2^2\,. \tag{35}$$

We have two cases.
First if $\|\tilde{\mathbf{w}} - \bar{\mathbf{w}}\|_3 \leq \frac{\epsilon}{9\,C_f}$. Consider the set $\mathcal{S} = \{\mathbf{w}\ :\ \|\mathbf{w} - \tilde{\mathbf{w}}\|_3 = \|\tilde{\mathbf{w}} - \bar{\mathbf{w}}\|_3\}$. For any $\mathbf{w}$ in the set $\mathcal{S}$ we have

$$\phi(\mathbf{b}, z, \mathbf{w}) - \phi(\mathbf{b}, z, \tilde{\mathbf{w}}) \geq \psi(\mathbf{b}, z, \mathbf{w}) - \psi(\mathbf{b}, z, \tilde{\mathbf{w}}) - \frac{C_f}{m}\left(\|\mathbf{w} - \bar{\mathbf{w}}\|_3^3 + \|\tilde{\mathbf{w}} - \bar{\mathbf{w}}\|_3^3\right)$$

$$\geq \frac{\epsilon}{m}\|\mathbf{w} - \tilde{\mathbf{w}}\|_2^2 - \frac{C_f}{m}\left(\|\mathbf{w} - \bar{\mathbf{w}}\|_3^3 + \|\tilde{\mathbf{w}} - \bar{\mathbf{w}}\|_3^3\right)$$

$$\geq \frac{\epsilon}{m}\|\mathbf{w} - \tilde{\mathbf{w}}\|_3^2 - \frac{C_f}{m}\left(4\|\mathbf{w} - \bar{\mathbf{w}}\|_3^3 + 5\|\tilde{\mathbf{w}} - \bar{\mathbf{w}}\|_3^3\right)$$

$$= \frac{9\,C_f}{m}\|\tilde{\mathbf{w}} - \bar{\mathbf{w}}\|_3^2\left(\frac{\epsilon}{9\,C_f} - \|\tilde{\mathbf{w}} - \bar{\mathbf{w}}\|_3\right) \geq 0\,. \tag{36}$$

This means that the optimal value of $\phi(\mathbf{b}, z, \mathbf{w})$ lies within $\mathcal{S}$. Now if $\mathbf{w}_\phi = \arg\min \phi(\mathbf{b}, z, \mathbf{w})$,

$$\Psi(\mathbf{b}, z) - \Phi(\mathbf{b}, z) = (\psi(\mathbf{b}, z, \tilde{\mathbf{w}}) - \psi(\mathbf{b}, z, \mathbf{w}_\phi)) + (\psi(\mathbf{b}, z, \mathbf{w}_\phi) - \phi(\mathbf{b}, z, \mathbf{w}_\phi))$$

$$\leq (\psi(\mathbf{b}, z, \mathbf{w}_\phi) - \phi(\mathbf{b}, z, \mathbf{w}_\phi)) \leq \frac{C_f}{m} \|\mathbf{w}_\phi - \bar{\mathbf{w}}\|_3^3$$

$$\leq \frac{4\,C_f}{m} \left( \|\mathbf{w}_\phi - \tilde{\mathbf{w}}\|_3^3 + \|\tilde{\mathbf{w}} - \bar{\mathbf{w}}\|_3^3 \right) \leq \frac{8\,C_f}{m} \|\tilde{\mathbf{w}} - \bar{\mathbf{w}}\|_3^3 . \tag{37}$$

And,

$$\Phi(\mathbf{b}, z) - \Psi(\mathbf{b}, z) = (\phi(\mathbf{b}, z, \mathbf{w}_\phi) - \phi(\mathbf{b}, z, \tilde{\mathbf{w}})) + (\phi(\mathbf{b}, z, \tilde{\mathbf{w}}) - \psi(\mathbf{b}, z, \tilde{\mathbf{w}}))$$

$$\leq (\phi(\mathbf{b}, z, \tilde{\mathbf{w}}) - \psi(\mathbf{b}, z, \tilde{\mathbf{w}})) \leq \frac{C_f}{m} \|\tilde{\mathbf{w}} - \bar{\mathbf{w}}\|_3^3 . \tag{38}$$

Thus, (**??**) and (**??**) implies that

$$|\Phi(\mathbf{b}, z) - \Psi(\mathbf{b}, z)| \leq \frac{8\,C_f}{m} \|\tilde{\mathbf{w}} - \bar{\mathbf{w}}\|_3^3 . \tag{39}$$

Case 2 if $\|\tilde{\mathbf{w}} - \bar{\mathbf{w}}\|_3 \geq \frac{\epsilon}{9\,C_f}$.

$$\Phi(\mathbf{b}, z) - \Psi(\mathbf{b}, z) = (\phi(\mathbf{b}, z, \mathbf{w}_\phi) - \phi(\mathbf{b}, z, \bar{\mathbf{w}})) + (\phi(\mathbf{b}, z, \bar{\mathbf{w}}) - \phi(\mathbf{0}, 0, \bar{\mathbf{w}}))$$

$$+ (\psi(\mathbf{0}, 0, \bar{\mathbf{w}}) - \psi(\mathbf{b}, z, \bar{\mathbf{w}})) + (\psi(\mathbf{b}, z, \bar{\mathbf{w}}) - \psi(\mathbf{b}, z, \tilde{\mathbf{w}}))$$

$$\leq \psi(\mathbf{b}, z, \bar{\mathbf{w}}) - \psi(\mathbf{b}, z, \tilde{\mathbf{w}}) \leq \frac{1}{2m} \left( z - \mathbf{b}^\mathsf{T}\bar{\mathbf{w}} \right)^2 . \tag{40}$$

$$\Psi(\mathbf{b}, z) - \Phi(\mathbf{b}, z) \leq (\psi(\mathbf{b}, z, \tilde{\mathbf{w}}) - \psi(\mathbf{b}, z, \bar{\mathbf{w}})) + (\psi(\mathbf{b}, z, \bar{\mathbf{w}}) - \psi(\mathbf{0}, 0, \bar{\mathbf{w}})) + (\phi(\mathbf{0}, 0, \bar{\mathbf{w}}) - \phi(\mathbf{0}, 0, \bar{\mathbf{w}}))$$

$$\leq \frac{1}{2m} \left( z - \mathbf{b}^\mathsf{T}\bar{\mathbf{w}} \right)^2 . \tag{41}$$

So finally,

$$|\Psi(\mathbf{b}, z) - \Phi(\mathbf{b}, z)| \leq \frac{1}{2m} \left( z - \mathbf{b}^\mathsf{T}\bar{\mathbf{w}} \right)^2 . \tag{42}$$

So by combining the two cases, we get

$$|\Phi(\mathbf{b}, z) - \Psi(\mathbf{b}, z)| \leq \mathbb{1}_{\|\tilde{\mathbf{w}} - \bar{\mathbf{w}}\|_3 \leq \frac{\epsilon}{9\,C_f}} \left( \frac{8\,C_f}{m} \|\tilde{\mathbf{w}} - \bar{\mathbf{w}}\|_3^3 \right) + \mathbb{1}_{\|\tilde{\mathbf{w}} - \bar{\mathbf{w}}\|_3 > \frac{\epsilon}{9\,C_f}} \left( \frac{1}{2m} \left( z - \mathbf{b}^\mathsf{T}\bar{\mathbf{w}} \right)^2 \right) . \tag{43}$$

Therefore,

$$\mathbb{E}\left[ |\Phi(\mathbf{b}, z) - \Psi(\mathbf{b}, z)| \right] \leq \mathbb{E}\left[ \mathbb{1}_{\|\tilde{\mathbf{w}} - \bar{\mathbf{w}}\|_3 \leq \frac{\epsilon}{9\,C_f}} \left( \frac{8\,C_f}{m} \|\tilde{\mathbf{w}} - \bar{\mathbf{w}}\|_3^3 \right) \right] + \mathbb{E}\left[ \mathbb{1}_{\|\tilde{\mathbf{w}} - \bar{\mathbf{w}}\|_3 > \frac{\epsilon}{9\,C_f}} \left( \frac{1}{2m} \left( z - \mathbf{b}^\mathsf{T}\bar{\mathbf{w}} \right)^2 \right) \right]$$

$$\leq \frac{8 C_f}{m} \mathbb{E}\left[ \|\tilde{\mathbf{w}} - \bar{\mathbf{w}}\|_3^3 \right] + \frac{1}{2m} \sqrt{\Pr\left\{ \|\tilde{\mathbf{w}} - \bar{\mathbf{w}}\|_3 \geq \frac{\epsilon}{9\,C_f} \right\} \mathbb{E}[(z - \mathbf{b}^\mathsf{T}\bar{\mathbf{w}})^4]}$$

$$\leq \frac{8 C_f}{m} \mathbb{E}\left[ \|\tilde{\mathbf{w}} - \bar{\mathbf{w}}\|_3^3 \right] + \frac{1}{2m} \sqrt{\frac{\mathbb{E}\left[ \|\tilde{\mathbf{w}} - \bar{\mathbf{w}}\|_3^3 \right]}{(\frac{\epsilon}{9 C_f})^3} \mathbb{E}[(z - \mathbf{b}^\mathsf{T}\bar{\mathbf{w}})^4]}$$

$$\leq \frac{C}{m^{5/4}} \tag{44}$$

On the other hand, it is easy to see that

$$\Psi(\mathbf{b}, z) - \Psi(\mathbf{0}, 0) = \frac{(z - \mathbf{b}^\mathsf{T}\bar{\mathbf{w}})^2}{2m(1 + \mathbf{b}^\mathsf{T}\Omega^{-1}\mathbf{b})} , \tag{45}$$

where $\Omega = \mathbf{V} + \mathbf{M}^\mathsf{T}\mathbf{M}$. Note that

$$\left| \mathbb{E}\left[ \Psi(\mathbf{b}, z) - \Psi(\mathbf{0}, 0) - \frac{\sigma^2 + \frac{\|\bar{\mathbf{w}}\|^2}{m}}{2m(1 + \mathbb{E}[\mathbf{b}^\mathsf{T}\Omega\mathbf{b}])} \right] \right| = \left| \mathbb{E}\left[ \frac{(z - \mathbf{b}^\mathsf{T}\bar{\mathbf{w}})^2}{2m(1 + \mathbf{b}^\mathsf{T}\Omega^{-1}\mathbf{b})} - \frac{\sigma^2 + \frac{\|\bar{\mathbf{w}}\|^2}{m}}{2m(1 + \mathbb{E}[\mathbf{b}^\mathsf{T}\Omega\mathbf{b}])} \right] \right|$$

$$\leq \frac{1}{2m} \mathbb{E}\left[ (z - \mathbf{b}^\mathsf{T}\bar{\mathbf{w}})^2 \left| \mathbf{b}^\mathsf{T}\Omega\mathbf{b} - \mathbb{E}[\mathbf{b}^\mathsf{T}\Omega\mathbf{b}] \right| \right]$$

$$\leq \frac{1}{2m} \sqrt{\mathbb{E}\left[ (z - \mathbf{b}^\mathsf{T}\bar{\mathbf{w}})^4 \right] \mathbb{E}\left[ (\mathbf{b}^\mathsf{T}\Omega\mathbf{b} - \mathbb{E}[\mathbf{b}^\mathsf{T}\Omega\mathbf{b}])^2 \right]}$$

$$\leq \frac{C}{m^{1+\tau/2}} . \tag{46}$$

Now putting (**??**) and (**??**) in (**??**), results in

$$\left| \mathbb{E}_{\{z,\mathbf{b}\}} \left[ \Phi(\mathbf{b}, z) - \bar{\Phi} - \frac{\sigma^2 + \frac{\|\bar{\mathbf{w}}\|^2}{m}}{2m(1 + \mathbb{E}[\mathbf{b}^\mathsf{T}\boldsymbol{\Omega}\mathbf{b}])} \right] \right| \leq \frac{8C_f}{m} \mathbb{E}\left[\|\tilde{\mathbf{w}} - \bar{\mathbf{w}}\|_3^3\right] + \frac{27C_f^{3/2}}{2m\epsilon^{3/2}} \sqrt{\mathbb{E}\left[\|\tilde{\mathbf{w}} - \bar{\mathbf{w}}\|_3^3\right] \mathbb{E}[(z - \mathbf{b}^\mathsf{T}\bar{\mathbf{w}})^4]}$$

$$+ \frac{1}{2m} \sqrt{\mathbb{E}\left[(z - \mathbf{b}^\mathsf{T}\bar{\mathbf{w}})^4\right] \mathbb{E}\left[(\mathbf{b}^\mathsf{T}\boldsymbol{\Omega}\mathbf{b} - \mathbb{E}[\mathbf{b}^\mathsf{T}\boldsymbol{\Omega}\mathbf{b}])^2\right]}$$

(47)

$$c\, m^{-(1+\tau/2)}$$

(48)

It remains to bound $\mathbb{E}\left[(z - \mathbf{b}^\mathsf{T}\bar{\mathbf{w}})^4\right]$ and $\mathbb{E}\left[\|\tilde{\mathbf{w}} - \bar{\mathbf{w}}\|_3^3\right]$. For the first one, let $\frac{1}{n}\mathbf{e} = \mathbb{E}[\mathbf{b}]$ and $\tilde{\mathbf{b}} = \mathbf{b} - \frac{1}{n}\mathbf{e}$. Then,

$$\mathbb{E}\left[(z - \mathbf{b}^\mathsf{T}\bar{\mathbf{w}})^4\right] = \mathbb{E}[z^4] + 6\mathbb{E}[z^2]\,\mathbb{E}[(\mathbf{b}^\mathsf{T}\bar{\mathbf{w}})^2] + \mathbb{E}[(\mathbf{b}^\mathsf{T}\bar{\mathbf{w}})^4]$$

$$= \mathbb{E}[z^4] + \frac{6\mathbb{E}[z^2]}{n}\left(\mathbb{E}[(\tilde{\mathbf{b}}^\mathsf{T}\bar{\mathbf{w}})^2] + (\mathbf{e}^\mathsf{T}\bar{\mathbf{w}})^2\right) + \mathbb{E}[(\tilde{\mathbf{b}}^\mathsf{T}\bar{\mathbf{w}})^4] + 6\mathbb{E}[(\tilde{\mathbf{b}}^\mathsf{T}\bar{\mathbf{w}})^2](\mathbf{e}^\mathsf{T}\bar{\mathbf{w}})^2 + (\mathbf{e}^\mathsf{T}\bar{\mathbf{w}})^4$$

$$\leq C_1 + C_2\|\bar{\mathbf{w}}\|^2 + C_3\|\bar{\mathbf{w}}\|^4 .$$

(49)

On the other hand, let $\boldsymbol{\Omega}^{-1} = [\omega_1 \ldots, \omega_n]^\mathsf{T}$. Since $\boldsymbol{\Omega}^{-1} \preceq 1/\epsilon$,

$$\mathbb{E}\left[\|\tilde{\mathbf{w}} - \bar{\mathbf{w}}\|_3^3\right] = \mathbb{E}\left[\left\|\frac{(z - \mathbf{b}^\mathsf{T}\bar{\mathbf{w}})}{(1 + \mathbf{b}^\mathsf{T}\boldsymbol{\Omega}^{-1}\mathbf{b})}\boldsymbol{\Omega}^{-1}\mathbf{b}\right\|_3^3\right] \leq \mathbb{E}\left[\left\|(z - \mathbf{b}^\mathsf{T}\bar{\mathbf{w}})\boldsymbol{\Omega}^{-1}\mathbf{b}\right\|_3^3\right]$$

$$\leq 4\,\mathbb{E}\left[\left\|(z - \mathbf{b}^\mathsf{T}\bar{\mathbf{w}})\boldsymbol{\Omega}^{-1}\tilde{\mathbf{b}}\right\|_3^3\right] + \frac{4}{n^3}\mathbb{E}\left[\left\|(z - \mathbf{b}^\mathsf{T}\bar{\mathbf{w}})\boldsymbol{\Omega}^{-1}\mathbf{e}\right\|_3^3\right]$$

$$\leq 4\sqrt{\mathbb{E}\left[(z - \mathbf{b}^\mathsf{T}\bar{\mathbf{w}})^6\right]\,\mathbb{E}\left[\left\|\boldsymbol{\Omega}^{-1}\tilde{\mathbf{b}}\right\|_3^6\right]} + \frac{4}{n^3}\left\|\boldsymbol{\Omega}^{-1}\mathbf{e}\right\|_3^3\sqrt{\mathbb{E}\left[(z - \mathbf{b}^\mathsf{T}\bar{\mathbf{w}})^6\right]}$$

$$\leq 4\sqrt{\mathbb{E}\left[(z - \mathbf{b}^\mathsf{T}\bar{\mathbf{w}})^6\right]\,\mathbb{E}\left[\sum_k |\omega_k^\mathsf{T}\tilde{\mathbf{b}}|^3\right]^2} + \frac{4}{n^3}\left\|\boldsymbol{\Omega}^{-1}\mathbf{e}\right\|_2^3\sqrt{\mathbb{E}\left[(z - \mathbf{b}^\mathsf{T}\bar{\mathbf{w}})^6\right]}$$

$$\leq \left(\frac{C}{n\epsilon^3} + \frac{4\|\mathbf{e}\|_2^3}{\epsilon^2 n^3}\right)\sqrt{\mathbb{E}\left[(z - \mathbf{b}^\mathsf{T}\bar{\mathbf{w}})^6\right]}$$

(50)

which concludes the proof. $\qquad\square$

**Theorem 4.** *let* $\mathbf{W_A}$ *and* $\mathbf{W_B}$ *bt the optimal solutions to* (**??**)*. If for any function* $f(.)$*, that satisfies our conditions,*

$$\Phi_\mathbf{A} - \Phi_\mathbf{B} \to 0 ,$$

(51)

*then,*

$$\frac{1}{n^2}\|\mathbf{W_A}\|_F^2 - \frac{1}{n^2}\|\mathbf{W_B}\|_F^2 \to 0 .$$

(52)

*Proof.* Assume that $\frac{1}{n^2}\|\mathbf{W_A}\|_F^2$ and $\frac{1}{n^2}\|\mathbf{W_B}\|_F^2$ converge to difference values of $C_\mathbf{A}$ and $C_\mathbf{B}$. Choose $C = (C_\mathbf{B} + C_\mathbf{A})/2$ and consider the following optimization,

$$\bar{\Phi}_\mathbf{A} = \min_{\substack{\frac{1}{n^2}\|\mathbf{W}\|_F^2 \leq C \\ \mathbf{W}\in\mathbb{H}^n}} \frac{1}{2m}\sum_{i=1}^m (z_i - \mathrm{Tr}(\mathbf{A}_i \cdot \mathbf{W}))^2 + f(\mathbf{W}) ,$$

$$\bar{\Phi}_\mathbf{B} = \min_{\substack{\frac{1}{n^2}\|\mathbf{W}\|_F^2 \leq C \\ \mathbf{W}\in\mathbb{H}^n}} \frac{1}{2m}\sum_{i=1}^m (z_i - \mathrm{Tr}(\mathbf{B}_i \cdot \mathbf{W}))^2 + f(\mathbf{W}) .$$

(53)

We show that the two should converge to the same value, which is a contradiction since $f(.)$ is strongly convex and one should converge to $\Phi_\mathbf{A}$ and the other should be larger that $\Phi_\mathbf{B}$. Using

min-max theorem, they can be rewritten as

$$\bar{\Phi}_{\mathbf{A}} = \sup_{\lambda > 0} \ -\lambda\, C + \min_{\mathbf{W} \in \mathbb{H}^n} \ \frac{1}{2m} \sum_{i=1}^{m} (z_i - \mathrm{Tr}(\mathbf{A}_i \cdot \mathbf{W}))^2 + f(\mathbf{W}) + \frac{\lambda}{n^2} \|\mathbf{W}\|_F^2 \,,$$

$$\bar{\Phi}_{\mathbf{B}} = \sup_{\lambda > 0} \ -\lambda\, C + \min_{\mathbf{W} \in \mathbb{H}^n} \ \frac{1}{2m} \sum_{i=1}^{m} (z_i - \mathrm{Tr}(\mathbf{B}_i \cdot \mathbf{W}))^2 + f(\mathbf{W}) + \frac{\lambda}{n^2} \|\mathbf{W}\|_F^2 \,. \tag{54}$$

Due to the assumption of the theorem, the two inside converge to the same value for any fixed $\lambda$. So the concave version of Lemma **??** shows that $\bar{\Phi}_{\mathbf{A}}$ and $\bar{\Phi}_{\mathbf{B}}$ also converge to the same value which is a contradiction. $\qquad\square$

**Lemma 1.** *Consider a series of convex functions $f_n : \mathbb{R}^{>0} \to \mathbb{R}$ that converges point-wise to the function $f : \mathbb{R}^{>0} \to \mathbb{R}$. Besides, there exists $M > 0$ such that for any $x > M$, we have $f(x) > \inf_{s>0} f(s)$. Then $f(.)$ is also convex and $\inf_{s>0} f_n(s) \xrightarrow{P} \inf_{s>0} f(s)$.*

**Lemma 2.** *Let $\bar{\mathbf{w}}$ be the optimal solution to the optimization*

$$\min_{\mathbf{w}} \ \frac{1}{2} \|\mathbf{z} - \mathbf{A}\mathbf{w}\|^2 + f(\mathbf{w} + \mathbf{x}_0) \,, \tag{55}$$

*where $f(.)$ is strongly convex with constant $\epsilon$. Then*

$$\|\bar{\mathbf{w}}\| \le \frac{2}{\epsilon} (\|\mathbf{A}^\top \mathbf{z}\| + \|\nabla f(\mathbf{x}_0)\|) \tag{56}$$

*Proof.* let

$$\phi(\mathbf{A}, \mathbf{w}) = \frac{1}{2} \|\mathbf{z} - \mathbf{A}\mathbf{w}\|^2 + f(\mathbf{w} + \mathbf{x}_0) \,. \tag{57}$$

We have

$$0 > \phi(\mathbf{A}, \bar{\mathbf{w}}) - \phi(\mathbf{A}, 0) \ge \bar{\mathbf{w}}^\top \left( -\mathbf{A}^\top \mathbf{z} + \nabla f(\mathbf{x}_0) \right) + \frac{\epsilon}{2} \|\bar{\mathbf{w}}\|^2 \,.$$

Therefore,

$$\frac{\epsilon}{2} \|\bar{\mathbf{w}}\|^2 \le \left| \bar{\mathbf{w}}^\top \left( -\mathbf{A}^\top \mathbf{z} + \nabla f(\mathbf{x}_0) \right) \right| \le \|\bar{\mathbf{w}}\| \ (\|\mathbf{A}^\top \mathbf{z}\| + \|\nabla f(\mathbf{x}_0)\|) \,, \tag{58}$$

which concludes the proof. Now let $\bar{\mathbf{w}}$ be the optimizer of $\phi(\mathbf{A}, \mathbf{w})$ and $\mathbb{E}[\mathbf{A}] = \mathbf{1}\mathbf{e}^\top$. Due to optimality we have,

$$0 = \mathbf{A}^\top (\mathbf{A}\bar{\mathbf{w}} - \mathbf{z}) + \nabla f(\mathbf{x}_0 + \bar{\mathbf{w}}) \tag{59}$$

$$\square$$

**Lemma 3.** *Let $\bar{\mathbf{w}}$ be the optimal solution to the optimization*

$$\min_{\mathbf{w}} \ \frac{1}{2} \|\mathbf{z} - \mathbf{A}\mathbf{w}\|^2 + f(\mathbf{w} + \mathbf{x}_0) \,, \tag{60}$$

*where $f(.)$ is strongly convex with constant $\epsilon$ and $\mathbf{A} \in \mathbb{R}^{m \times n}$ is a random value with $\mathbb{E}[\mathbf{A}] = \mathbf{1}\mathbf{e}^t$ and $\mathbf{B} = \mathbf{A} - \mathbf{1}\mathbf{e}^t$. Then*

$$\|\bar{\mathbf{w}}\| \le \frac{2}{\epsilon} (\|\mathbf{A}^\top \mathbf{z}\| + \|\nabla f(\mathbf{x}_0)\|) \tag{61}$$

*Proof.* We have,

$$\phi(\mathbf{A}, 0) \ge \phi(\mathbf{A}, \bar{\mathbf{w}}) = \frac{1}{2} \|\mathbf{z} - \mathbf{B}\bar{\mathbf{w}} - \mathbf{1}\mathbf{e}^\top \bar{\mathbf{w}}\|^2 + f(\bar{\mathbf{w}} + \mathbf{x}_0) \ge \frac{m}{2} (\mathbf{e}^\top \bar{\mathbf{w}})^2 + (\mathbf{e}^\top \bar{\mathbf{w}}) \cdot \mathbf{1}^\top (\mathbf{B}\bar{\mathbf{w}} - \mathbf{z}) \tag{62}$$

Therefore,

$$(\mathbf{e}^\top \bar{\mathbf{w}})^2 + \frac{2}{m} (\mathbf{e}^\top \bar{\mathbf{w}}) \cdot \mathbf{1}^\top (\mathbf{B}\bar{\mathbf{w}} - \mathbf{z}) - \frac{2}{m} \phi(\mathbf{A}, 0) \le 0 \tag{63}$$

This results in

$$|\mathbf{e}^\top \bar{\mathbf{w}}| \le \frac{2}{m} \left| \mathbf{1}^\top (\mathbf{B}\bar{\mathbf{w}} - \mathbf{z}) \right| + \frac{2}{m} \phi(\mathbf{A}, 0) \le \frac{2}{m} |\mathbf{1}^\top \mathbf{z}| + \frac{2}{m} \|\bar{\mathbf{w}}\| \cdot \|\mathbf{B}^\top \mathbf{1}\| + \frac{2}{m} \|\mathbf{z}\|^2 + f(\mathbf{x}_0)$$

$$\le \frac{2}{m} |\mathbf{1}^\top \mathbf{z}| + \frac{4}{m\epsilon} (\|\mathbf{A}^\top \mathbf{z}\| + \|\nabla f(\mathbf{x}_0)\|) \cdot \|\mathbf{B}^\top \mathbf{1}\| + \frac{2}{m} \|\mathbf{z}\|^2 + f(\mathbf{x}_0) \tag{64}$$

$$\square$$