[Reviews · NeurIPS 2019]

Reviewer 1



The paper should provide a more thorough comparison with existing results on the characterizations of non-Gaussian random matrices (e.g., binary, Rademacher, etc.). In particular, whether the analytical results in this paper meet or exceed the performance guarantees already available in the literature for several non-Gaussian matrix designs. (This has been addressed in the rebuttal) Clarity: * The statement in lines 51-55 does not discuss any dependence of the number of measurements on the signal observed x_0. * The corollaries (1,2,...) should be linked to the Theorems from CGMT to which they are derived from. * The statements in lines 233-234 ("resembles the definition of the Gaussian width") should be more precise. Is (5) the Gaussian width? Or are there differences, and what are they? (This has been addressed in the rebuttal) * In line 247, should the matrix M be normalized so that its Gramian could match the covariance matrix? * In line 307, "Corollary 3 indicates that 3rn measurements...": the result in the corollary is asymptotic, while this statement appears to describe a finite value of n; so the two statements do not seem to be equivalent? (This has been addressed in the rebuttal) * The supplemental material has many broken references (??) Minor items and typos: Line 39 define acronym iid Line 59 vec(X) should be vec(X_0) Line 64 should non-negativity be positive (semi)definiteness? Line 67 et al -> et al. Line 185 funtion -> function Lines 251, 253 i.i.d -> i.i.d. Line 272 There is no need to define the assumption within the assumption. Line 301 the later -> the latter

Reviewer 2



Overall a very nice paper. I don't have many comments, as I found the results quite interesting and the paper well-written. It is quite relevant to the community as it enables a wider range of matrices to be used. === I've seen and taken into account the author's response, and it does not change my score.

Reviewer 3



The results are of a high quality and certainly deserving of publication. The results are original, but not particularly surprising or timely. There has been an intense study of convex relaxation for recovery of structured signals, with the initial optimal order theoretical results starting in approximately 2005 and 2006. The initial results were primarily for gaussian or randomised Fourier measurements. There has been a number of results showing that the results hold beyond these cases; some of which has been well cited, but others such as "Universality in polytope phase transitions and message passing algorithms" by Bayati. Lelarge, and Montanari is missing. Extending these minimal sampling bounds to a broader class of linear measurements is important, but not especially surprising and as a result I wonder if NeurIPS is the most suitable venue. While very pleased with the results, I expect that even for subject area experts in sparsity attending NeurIPS, that they would likely prefer to attend talk on more unexpected or timely topics. Outside this issue of innovation, the writing is clear and generally scholarly. That said there are a few areas that would benefit from minor improvements. Title: The title is not sufficiently informative as the results are for convex objectives, while the title does not convey this important restriction. The title should be adapted to reflect this restriction. Non-convex alternatives: There are now a number of computationally efficient non-convex algorithms with recovery guarantees under the RIP conditions. While theoretical these methods are not well understood, they do have good performance, especially for the low rank approximation problem where for small sampling rates they appear to circumvent the necessary factor of 3 in the phase transition for convex algorithms. Historical context: The results focus on deriving the asymptotic minimal sampling complexity. The thresholds presented were first derived in a series of papers by Donoho and Donoho and Tanner. Unfortunately these initial results are not cited, instead later results extending these original results are stated. It would be appropriate to cite these foundational works, such as the paper "Counting faces of randomly-projected polytopes when the projection radically lowers dimension" by Donoho and Tanner which originally proved the results 2k\log(n/k) stated on the top of page 2 and attributed to [12]. Moreover, if interested in notions of universality the authors might be interested in the paper "Counting the faces of randomly-projected hypercubes and orthants, with applications" by Donoho and Tanner where they prove similar sampling complexity that is truly universal in that it holds even for deterministic matrices.

[Author Response · NeurIPS 2019]

We would like to thank the reviewers for their time, constructive feedback, and critical suggestions that will help us improve the paper. As noted in the reviews, the final submission of the paper needs additional refinements, especially in the proof sections presented in the supplementary material. Before providing our individual reponses, we would like to bring two important points to the reviewers' attentions. First, the theoretical results provided in this paper have many potential applications. Due to space limitations, we confined ourselves to only a few applications in learning from quadratic measurements, i.e., sparse covariance estimation, low-rank matrix recovery, and PhaseLift in phase retrieval. Exploring all potential applications of our main result (Theorem 1) is beyond the scope of a single conference paper. As a result, we anticipate that our theoretical findings may be helpful to many researchers and, in particular, to the NeurIPS audience. Second, we must emphasize that, unlike most previous works, in this paper we do not assume the independence of the entries of the measurement vectors. Indeed, to the best of our knowledge, the precise phase transition for measurements vectors with non i.i.d. entries has been provided for the first time in this paper. Please find our individual responses to each of the reviews in their respective threads.

**Reviewer1:** This paper provides a generalized framework from which the precise performance of signal reconstruction using linear measurements with iid vectors can be characterized. The importance of the result is that it extends the performance analysis beyond the iid Gaussian setting (which has been studied extensively in the literature). We thank the reviewer for the detailed comments and suggested improvements. Nonetheless, we must point out that previous results for non-Gaussian random matrices (e.g., binary or Rademacher) either have assumed that the entries of the measurement vectors are independent (and demonstrate a performance identical to the iid Gaussian case), or only provide orderwise bounds on the phase transition (such as those that have been given for PhaseLift). Our main result (Theorem 1) precisely characterizes the performance without assuming independence of the entries (which is why we were able to resolve the long-standing problem of determining the phase transition of PhaseLift). However, to address the reviewer's point, in the final submission we will add a discussion to compare with previous results for non-Gaussian random matrices. The statement in line 51-55 is informal and was meant to provide the gist of the main contribution. To add clarity, we will edit the text and more formally illustrate the result.

We will add some references to specific parts in the corresponding papers (e.g. Section $V$.C in [29]) to address the connections of our corollaries with the previous results exploiting CGMT. The remark on the "Pseudo Gaussian Width" will be updated to avoid ambiguities. The expectation defined in (5), is a generalization of the Gaussian width that is known in the literature. More specifically, as noted in the paper, it reduces to the Gausssian width in the iid Gaussian setting, i.e., when $\mu = 0$, and $\Sigma = \mathbf{I}_n$. In line 247, the matrix $\mathbf{M}$ is only used for generating a PSD covariance matrix through $\Sigma = \mathbf{M}\mathbf{M}^T$, where $\Sigma$ is used later on in our numerical simulations. Therefore, normalization is not needed. The reviewer is correct that the result in Corollary 3 is assymptotic, based on which we should be more careful in our statement in line 307 (we will have perfect recovery if $m/n > 3r$). It is worth noting that even though the theoretical results are asymptotic, we observed in our numerical simulations that when $n$ and $m$ are moderately large, the theory well matches the empirical results.

We appreciate the typographical comments from the reviewer. We also apologize for the broken references in the supplementary material. They will all be taken care of in the final submission of the paper.

**Reviewer2:** As correctly pointed out by the reviewer, this paper generalizes results in linear inverse problems to beyond iid Gaussian settings. The main theoretical result (in Theorem 1) demonstrates the equivalence of the phase transition in the Gaussian and non-Gaussian settings provided the first and second order statistics match. We provide detailed specialization of the result for sparse and low-rank matrix recovery, and PhaseLift in phase retrieval. We are glad to see the reviewer finds the results interesting and widely applicable. We also thank the reviewer for the suggested improvements. We will provide some clarification for the definition of "pseudo Gaussian width". We will also add a section to discuss potential further research directions and suggest some other cases where our theory can be applied.

**Reviewer3:** As stated by the reviewer, this paper extends earlier results on the performance of convex-relaxation-based methods to a broader class of linear measurements. We appreciate the efforts made by the reviewer to provide us with the insightful historical context. The comments are definitely valid and will be addressed in the final submission of the paper. We are planning to extend the introduction section by (a) adding a detailed discussion on the foundational works that analyzed the performance of convex relaxations for solving the linear inverse problem, and (b) a comparison with other universality results which have especially gained attention in the community in the past few years.

Theorem 1 was developed as an effort to resolve some of the current trending topics in phase retrieval, learning from quadratic measurements, interference analysis in MIMO communications, etc., for which the classical results could not provide a precise analysis. Regarding the comments on alternative non-convex methods, as the reviewer points out, the theoretical understanding is very limited. As such, it is not straightforward—and certainly beyond the scope of this paper—to compare our theoretical findings with the plethora of available non-convex methods. Nonetheless, we are currently investigating whether (under certain conditions) the phase transition of the optimization program (3) with non-convex objective $f(\cdot)$, would remain the same if we replace the rows of the measurement matrix, $\mathbf{A}$, with vectors drawn from Gaussian distribution with the same first and second moments. Finally, at this point we feel that the abstract of the paper sufficiently describes the convex limitation of our results. However, if the meta-reviewer feels that this restriction should also be reflected in the title, we will happily do so.

[Meta-Review · NeurIPS 2019]

The paper studies the minimum number of linear measurements requires to recover a sparse, low rank, or otherwise structured signal when using convex relaxations. The key contribution is the extension of the class of linear measurements designs by requiring only the first and second moments of the measurement ensemble. This is an important extension. The reviewers and also I recommend a modification in the title since "learning" in all generality is not proved in the manuscript but rather the convex formulations are. As a result, it would be important to highlight this distinction.